# A Learning Based Hypothesis Test for Harmful Covariate Shift

**Tom Ginsberg & Zhongyuan Liang & Rahul G. Krishnan**
Department of Computer Science
University of Toronto
Toronto, ON M5S 1A1
{tomginsberg,zhongyuan,rahulgk}@cs.toronto.edu

## Abstract

The ability to quickly and accurately identify covariate shift at test time is a critical and often overlooked component of safe machine learning systems deployed in high-risk domains. While methods exist for detecting when predictions should not be made on out-of-distribution test examples, identifying distributional level differences between training and test time can help determine when a model should be removed from the deployment setting and retrained. In this work, we define *harmful covariate shift* (HCS) as a change in distribution that may weaken the generalization of a predictive model. To detect HCS, we use the discordance between an ensemble of classifiers trained to agree on training data and disagree on test data. We derive a loss function for training this ensemble and show that the disagreement rate and entropy represent powerful discriminative statistics for HCS. Empirically, we demonstrate the ability of our method to detect harmful covariate shift with statistical certainty on a variety of high-dimensional datasets. Across numerous domains and modalities, we show state-of-the-art performance compared to existing methods, particularly when the number of observed test samples is small[1].

## 1 Introduction

Machine learning models operate on the assumption, albeit incorrectly that they will be deployed on data distributed identically to what they were trained on. The violation of this assumption is known as distribution shift and can often result in significant degradation of performance [Bickel et al., 2009; Rabanser et al., 2019; Otles et al., 2021; Ovadia et al., 2019]. There are several cases where a mismatch between training and deployment data results in very real consequences on human beings. In healthcare, machine learning models have been deployed for predicting the likelihood of sepsis. Yet, as [Habib et al., 2021] show, such models can be miscalibrated for large groups of individuals, directly affecting the quality of care they experience. The deployment of classifiers in the criminal justice system [Hao, 2019], hiring and recruitment pipelines [Dastin, 2018] and self-driving cars [Smiley, 2022] have all seen humans affected by the failures of learning models. The need for methods that quickly detect, characterize and respond to distribution shift is, therefore, a fundamental problem in trustworthy machine learning. We study a special case of distribution shift, commonly known as *covariate shift*, which considers shifts only in the distribution of input data $\mathbb{P}(X)$ while the relation between the inputs and outputs $\mathbb{P}(Y|X)$ remains fixed. In a standard deployment setting where ground truth labels are not available, covariate shift is the only type of distribution shift that can be identified.

For practitioners, regulatory agencies and individuals to have faith in deployed predictive models without the need for laborious manual audits, we need methods for the identification of covariate shift that are *sample-efficient* (identifying shifts from a small number of samples), *informed* (identifying shifts relevant to the domain and learning algorithm), *model-agnostic* (identifying shifts regardless of the functional class of the predictive model) and *statistically sound* (identifying true shifts while avoiding false positives with high-confidence).

---

[1] Code available at https://github.com/rgklab/detectron

We build off recent progress in understanding model performance under covariate shift using the PQ-learning framework [Goldwasser et al., 2020], a framework for selective classifiers that may either predict on or reject a given sample, that provides strong performance guarantees on arbitrary test distributions. Our work uses and extends PQ-learning to develop a practical, model-based hypothesis test, named *the Detectron*, to identify potentially harmful covariate shifts given any existing classification model already in deployment.

Our work makes the following key contributions:

- We show how to construct an ensemble of classifiers that maximize out-of-domain disagreement while behaving consistently in the training domain. We propose the *disagreement cross entropy* for models learned via continuous gradient-based methods (e.g., neural networks), as well as a generalization for those learned via discrete optimization (e.g., random forest).
- We show that the rejection rate and the entropy of the learning ensemble can be used to define a model-aware hypothesis test for covariate shift, the Detectron, that in idealized settings can provably detect covariate shift.
- On high-dimensional image and tabular data, using both neural networks and gradient boosted decision trees, our method outperforms state-of-the-art techniques for detecting covariate shift, particularly when given access to as few as ten test examples.

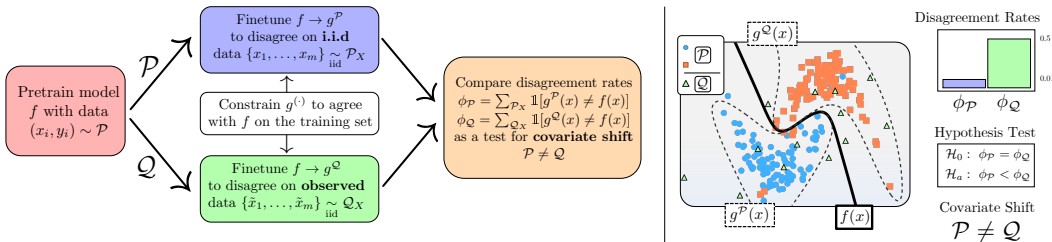

Figure 1: **Overview of the Detectron:** Starting with a base classifier $f$ trained on labeled samples from distribution $\mathcal{P}$ we train new *Constrained Disagreement Classifiers* (CDCs) $g^{\mathcal{P}}$ and $g^{\mathcal{Q}}$ on a small sets of unseen samples from $\mathcal{P}$ as well as a unknown distribution $\mathcal{Q}$. CDCs aim to maximize classification disagreement on unseen data while constrained to classify consistently with $f$ on their original training set. The rate $\phi$ that CDCs disagree is a powerful and sample efficient statistic for identifying covariate shift $\mathcal{P} \neq \mathcal{Q}$.

## 2 BACKGROUND AND RELATED WORK

**Covariate Shift Detection.** Covariate shift is the tendency for a distribution at test time $p_{\text{test}}(x)$ to differ from that seen during training $p_{\text{train}}(x)$ while the underlying prediction concept $y$ remains fixed e.g. $p_{\text{train}}(y|x) = p_{\text{test}}(y|x)$. Many methods for detecting shift apply dimensionality reduction followed by statistical hypothesis tests for distributional differences in the outputs (from a reference and target) [Rabanser et al., 2019]. Rabanser et al. show that using the softmax outputs of a pretrained classifier as low dimensional representations for performing univariate KS-tests, a method known as black box shift detection (BBSD) [Lipton et al., 2018], is effective at confidently identifying several synthetic covariate shifts in imaging data (e.g. crops, rotations) given approximately 200 i.i.d samples. However, applying statistical tests to non-invertible representations of data can never guarantee to capture arbitrary covariate shifts, as there may always exist multiple distributions that collapse to the same test statistic [Zhang et al., 2021]. Kifer et al. [2004]; Ben-David et al. [2006] introduce some of the earliest learning theoretic approaches for identifying and correcting for covariate shift based on discriminative learning with finite samples. More recent approaches for covariate shift detection including classifier two sample tests [Lopez-Paz and Oquab, 2017], deep kernel MMD [Liu et al., 2020] and H-Divergence [Zhao et al., 2022] rely on analyzing the outputs of unsupervised learning models (see Appendix subsection E.3 for more details). In our work we take a transductive learning approach and construct a method to directly use the structure of a supervised classification problem to improve the statistical power for detecting shifts.

**Out of Distribution Detection.** Out of distribution (OOD) detection focuses on identifying when a specific data point $x'$ admits low likelihood under the original training distributions ($p_{\text{train}}(x') \approx 0$)— a useful tool to have at inference time. Ren et al. [2019]; Morningstar et al. [2021] represent

a broad class of work that uses density estimation to pose the identification of covariate shift as anomaly detection. Others, including ODIN [Liang et al., 2018], Deep Mahalanobis Detectors [Lee et al., 2018] and, Gram Matrices [Sastry and Oore, 2020] directly use the predictive model (e.g. information from the intermediate representations of neural networks). The majority of modern methods in this space have been designed exclusively for deep neural networks, an uncommon modelling choice particularly for tabular data [Borisov et al., 2021]. Related to OOD, is the task of estimating sources of uncertainty in model predictions [Lakshminarayanan et al., 2017; Ovadia et al., 2019]. Naturally, uncertainty should be large when samples are OOD, however [Ovadia et al., 2019] perform a large-scale empirical comparison of uncertainty estimation methods and find that while deep ensembles generally provide the best results, the quality of uncertainty estimations, regardless of method, consistently degrades with increasing covariate shift.

**Selective Classification and PQ Learning.** Selective classification concerns building classifiers that may either predict on or reject on test samples [Geifman and El-Yaniv, 2019]. Recent work by [Goldwasser et al., 2020] develops a formal framework known as PQ learning which extends probably approximately correct (PAC) learning [Haussler, 1990] to arbitrary test distributions by allowing for selective classification. While PAC learning concerns the development of a classifier with a bounded finite-sample error rate on its training distribution, PQ learning seeks a selective classifier with jointly bounded finite-sample error and rejection rates on arbitrary test distributions. The Rejectron algorithms proposed therein builds an ensemble of models that produce different outputs relative to a perfect baseline on a set of unlabeled test samples. We provide a summary of the original Rejectron algorithm in the supplementary material (see Appendix A). PQ-learning represents a major theoretical leap for learning guarantees under covariate shift; however, the majority of the underlying ideas have not been implemented/tested experimentally using real-world data. We show how to build a PQ learner by generalizing the Rejectron algorithm, overcoming several limitations and assumptions made by the original work including extending beyond simple binary classification to general multiclass and multilabel tasks and reducing the number of samples required for learning at each iteration.

## 3 DETECTRON

**Problem Setup.** Let $f : X \to Y$ be classification model from a function class $F$ that maps from space of covariates $X$ to a discrete set of classes $Y = \{1, \ldots, N\}$. We assume $f$ was trained on a dataset of labeled samples $\mathbf{P} = \{(x_i, y_i)\}_{i=1}^{n}$ where each $x_i$ is drawn identically from a distribution $\mathcal{P}$ over $X$. In deployment, $f$ is then made to predict on new unlabeled samples $\mathbf{Q} = \{\tilde{x}_i\}_{i=1}^{m}$ from a distribution $\mathcal{Q}$ over $X$. Our goal is to determine whether $f$ may be trusted to do so accurately. The problem we address is how to automatically detect, from only a set of finite samples $\mathbf{Q}$, if the new covariate distribution $\mathcal{Q}$ has shifted from $\mathcal{P}$ in such a way that $f$ can no longer be assumed to generalize — we refer to this type of dataset shift as *harmful covariate shift*.

**Harmful Covariate Shift.** A shift in the data distribution is not always harmful. In many practical problems, a practitioner may use domain knowledge to embed invariances with the explicit goal of ensuring the predictive performance of a classifier does not, by construction, change under certain shifts. This may be done directly via translation invariance in convolutional neural networks, permutation invariance in DeepSets [Zaheer et al., 2017] or indirectly via data augmentation or domain adaptation. Such practical heuristics can lead to models generalizing to a more broad range of distributions than can be characterized by just the training set. We refer to such an induced generalization set as $\mathcal{R}$. Although $\mathcal{R}$ is difficult to characterize and will in general depend on the model architecture, learning algorithm and training dataset, we seek a practical method for detecting shift that is explicitly tied to $\mathcal{R}^2$. We present a more formal definition of *harmful covariate shift* as well as a connection to domain identification and $\mathcal{A}$ distance [Kifer et al., 2004] in Appendix H.

Our approach is based both on PQ learning and intuition from learning theory. If there exists a set of classifiers with the same generalization set $\mathcal{R}$ but behave inconsistently on samples from a distribution $\mathcal{Q}$, then $\mathcal{Q}$ must not be a member $\mathcal{R}$. Our strategy will be to create an ensemble of *constrained disagreement classifiers*, classifiers constrained to predict consistently (i.e., predict the same as $f$) on $\mathcal{R}$ but as differently as possible on $\mathcal{Q}$. If $\mathcal{Q}$ is within $\mathcal{R}$ then such an ensemble will fail to predict differently. When we can find an ensemble that exhibits inconsistent behaviour on $\mathcal{Q}$, there

---

[2] See Appendix G for further intuition and a concrete example of the generalization set.

must be covariate shift that explicitly lies outside $\mathcal{R}$. To make the idea of constrained disagreement classifiers tangible we propose a simple definition which we will translate into a learning algorithm in the following sections.

**Constrained Disagreement Classifier (CDC).** A constrained disagreement classifier $g_{(f,\mathbf{P},\mathbf{Q})}$, or simply $g$ if $f$, $\mathbf{P}$ and $\mathbf{Q}$ are clear with context, is a classifier with the following properties:

1. $g$ belongs to the same model class as $f \in F$ and is trained with the same algorithm on $\mathbf{P}$
2. $g$ achieves similar performance on unseen samples drawn i.i.d to $\mathbf{P}$
3. $g$ disagrees maximally with $f$ on elements of dataset $\mathbf{Q}$ while not violating 1 and 2

Our definition of a CDC aims to explicitly capture the concept of a classifier that learns the same generalization region as $f$ while behaving as inconsistently as possible on $\mathbf{Q}^3$.

**Limitations of PQ Learning.** As our work builds on PQ learning, we provide a summary of the original framework and clearly state the distinctions in our methodology. In PQ learning we seek a selective classifier $h$ that achieves a bounded tradeoff between its in distribution rejection rate $\mathrm{rej}_h(\mathbf{x})$ and its out of distribution error $\mathrm{err}_h(\tilde{\mathbf{x}})$ with respect to a ground truth decision function $d$. Formally, this tradeoff is defined using the following learning theoretic bound (an extended description can be found in Appendix A.1).

*PQ learning [Goldwasser et al., 2020]* Learner $L$ $(\epsilon, \delta, n)$-PQ-learns $F$ if for any distributions $\mathcal{P}, \mathcal{Q}$ over $X$ and any ground truth function $d \in F$, its output $h := L(\mathbf{P}, d(\mathbf{P}), \mathbf{Q})$ satisfies

$$\mathbb{P}_{\mathbf{x} \sim \mathcal{P}^n, \, \tilde{\mathbf{x}} \sim \mathcal{Q}^n} \left[ \mathrm{rej}_h(\mathbf{x}) + \mathrm{err}_h(\tilde{\mathbf{x}}) \leq \epsilon \right] \geq 1 - \delta \tag{1}$$

Goldwasser et al. propose the Rejectron algorithm for PQ learning (summarized in Appendix A) in a noiseless binary classification setting with zero training error and access to a perfect empirical risk minimization oracle. We highlight several pitfalls that prevent a practical realization of Rejectron (1) the classification problem must be binary (2) the optimization objective at each step requires an ERM query with $\Omega(|\mathbf{P}|^2)$ samples and (3) most significantly there is no process to control overfitting. By tackling the above issues we derive a practical PQ learner and propose a powerful hypothesis test to identify harmful covariate shift.

**Learning to Disagree with the Disagreement Cross Entropy.** To train a classifier to disagree in the binary setting, it suffices to flip the labels. However, in the multi-class classification, it is unclear what a good objective function is. We formulate an explicit loss function that can be minimized via gradient descent to learn a CDC. For classification problems, letting $\hat{y} := g(x_i)$ be the predictive distribution over $N$ classes with the $i^{\text{th}}$ denoted by $\hat{y}_i$, $f(x_i) \in \{1, \dots N\}$ the label predicted by $f$ and $\mathbb{1}_{(.)}$ a binary indicator, we define the *disagreement-cross-entropy* (DCE) $\tilde{\ell}$ as:

$$\tilde{\ell}(\hat{y}, f(x_i)) = \frac{1}{1 - N} \sum_{c=1}^{N} \mathbb{1}_{f(x_i) \neq c} \log(\hat{y}_c) \tag{2}$$

$\tilde{\ell}$ corresponds to taking the cross entropy of $\hat{y}$ with the uniform distribution over all classes except $f(x_i)$. Since the primary criteria is that $g(x_i)$ disagrees with $f(x_i)$, $\tilde{\ell}$ is designed to minimize the probability of $g$'s prediction for the output of $f$'s while maximizing its overall entropy. Our definition of $\tilde{\ell}$ is stable to optimize, has a bounded global minimum, and can be shown to have desirable properties for disagreement learning, see Appendix D for more details.

Our goal is to agree on $\mathbf{P}$ and disagree on $\mathbf{Q}$. Consequently, we learn with the loss in Equation 3. $\ell$ denotes the standard cross entropy loss and $\tilde{\ell}$ is the disagreement cross entropy. $\lambda$ is a scalar parameter that controls the trade off between agreement and disagreement.

$$\mathcal{L}_{\mathrm{CDC}}(\mathbf{P}, \mathbf{Q}) = \frac{1}{|\mathbf{P} \cup \mathbf{Q}|} \left( \sum_{(x_i, y_i) \in \mathbf{P}} \ell(g(x_i), y_i) + \lambda \sum_{\tilde{x}_i \in \mathbf{Q}} \tilde{\ell}(g(\tilde{x}_i), f(\tilde{x}_i)) \right) \tag{3}$$

When learning CDCs in practice, $\mathcal{L}_{\mathrm{CDC}}$ should be combined with any additional regularization and data augmentation used in the original training process of $f$ to ensure that we retain the true

---

[3] A discussion on the relation between model complexity of $F$ and the behaviour of CDCs and associated limitations of our defintion is given in Appendix I.

generalization region of $f$. Furthermore, training and validation metrics must be closely monitored on unseen samples from $\mathcal{P}$ to ensure that $g$ achieves similar generalization performance on $\mathcal{P}$.

**Choosing $\lambda$.** In the original formulation of Rejectron, selective classifiers are trained on a dataset consisting of $\mathbf{P}$ replicated $|\mathbf{P}|$ times and $\mathbf{Q}$. Calling an ERM oracle on this data ensures that a misclassification on $\mathbf{P}$ is significantly more costly than one on $\mathbf{Q}$ but requires $\Omega(|\mathbf{P}|^2)$ samples, an impractical number for large datasets. We show that instead we can choose the scalar parameter $\lambda$ in Equation 3 to set learning $\mathbf{P}$ as the primary learning objective and only when it cannot be improved, we allow $g$ to learn how to disagree on $\mathbf{Q}$. The reasoning is a simple counting argument. Suppose agreeing on each sample in $\mathbf{P}$ incurs a reward of 1 and disagreeing with each sample in $\mathbf{Q}$ a reward of $\lambda$. To encourage agreement on $\mathbf{P}$ as the primary objective, we set $\lambda$ such that the extra reward obtained by going from *zero* to *all* disagreements on $\mathbf{Q}$ is less than that achieved with only one extra agreement on $\mathbf{P}$, this gives $\lambda|\mathbf{Q}| < 1$. Practically, we chose $\lambda = 1/(|\mathbf{Q}| + 1)$ and find that no tuning is required.

Some predictive models require learning with non-differentiable loss functions and/or discrete optimization (e.g., random forest). In such cases we fall back and formulate a general disagreement loss by duplicating each sample in $\mathbf{Q}$ $(N - 1)$ times, giving each a unique label that is not the target and weight of $1/(N - 1)$. In the continuous case, this corresponds exactly to Equation 2.

To learn richer disagreement rules, we create an ensemble of CDCs where the $k^{\text{th}}$ model is trained only to disagree on the subset of $\mathbf{Q}$ that has yet to be disagreed on by models 1 through $k - 1$. The final disagreement rate $\phi_{\mathbf{Q}}$ is the fraction of unlabelled samples where any CDC provides an alternate decision from $f$. In what follows we use this rate to characterize shift.

**From Constrained Disagreement to Detecting Shift with Hypothesis Tests.** A natural way to apply the concept of constrained disagreement to the identification of covariate shift is to partition $\mathbf{Q}$ into two sets, using the first to train a CDC ensemble and the second to compute an unbiased estimate of its held out disagreement rate $\phi_{\mathcal{Q}}$. We would statistically compare this disagreement rate using a $2 \times 2$ exact hypothesis test against a baseline estimate for the disagreement rate on $\mathcal{P}$. The following shows that this results in a provably correct method to detect shift.

**Theorem 1** (Disagreement implies covariate shift). Let $f$ be a classifier trained on dataset $\mathbf{P}$ consisting of samples drawn identically from $\mathcal{P}$ and their corresponding labels. Let $g$ be a classifier that is observed to agree (classify identically) with $f$ on $\mathbf{P}$ and disagree on a dataset $\mathbf{Q}$ drawn from $\mathcal{Q}$. If the rate which $g$ disagrees with $f$ on $n$ unseen samples from $\mathcal{Q}$ is greater than that from $n$ unseen samples from $\mathcal{P}$ w.p. greater than $p^\star := \frac{1}{2}\left(1 - 4^{-n}\binom{2n}{n}\right)$ there must be covariate shift.

*Sketch of Proof.* We show that under the null hypothesis where $\mathcal{P} = \mathcal{Q}$ the tightest upper bound on the probability that $g$ is more likely to disagree on $\mathcal{Q}$ compared to $\mathcal{P}$ is $p^\star$. The contrapositive argument then states if we deem the probability to be greater than $p^\star$ there must be a covariate shift. This result motivates a hypothesis testing approach to determine how probable it is that $g$ is truly more likely to disagree on $\mathcal{Q}$ given only a set of finite observations. The full proof can be found in Appendix C.

Our theory, while simple, has a limitation that prevents its direct application. Any approach that requires unseen samples from $\mathbf{Q}$ is ill-suited for the low data regime, as it requires splitting $\mathbf{Q}$ leaving an even smaller set for computing the disagreement rate. Estimators from small samples result in high variance and ultimately low statistical power. Since our objective is to detect covariate shift from as few test samples as possible, splitting $\mathbf{Q}$ is not a good option. To tackle this issue practically, we take a transductive approach based on intuition from learning theory: creating learning models to disagree on samples from $\mathbf{Q}$ while generalizing to $\mathcal{R}$ is a far easier task when $\mathcal{Q}$ is not in $\mathcal{R}$. We can therefore use the *relative increase* in disagreement between CDCs on $\mathbf{Q}$ and $\mathbf{P}$ to capture a quantity that is nearly as informative as the unbiased statistic without reducing samples from $\mathbf{Q}$ that we can use.

**The Detectron Test.** Our proposed method is to train two CDC ensembles $g_{\mathbf{Q}}$ and $g_{\mathbf{P}}$. $g_{\mathbf{Q}}$ is trained to disagree on **all** of $\mathbf{Q}$ and a $g_{\mathbf{P}}$ is a baseline trained to disagree an unseen set $\mathbf{P}^\star$ from $\mathcal{P}$. Once trained, we compute the ratio of samples $\phi_{\mathbf{Q}}$ and $\phi_{\mathbf{P}}$ that $g_{\mathbf{Q}}$ and $g_{\mathbf{P}}$ learn to disagree on their sets respectively. Under the null hypothesis where $\mathcal{Q}$ belongs to the generalization region of $f$ and $\mathcal{P}$, $\mathbb{E}[\phi_{\mathbf{Q}}] \leq \mathbb{E}[\phi_{\mathbf{P}}]$, while harmful shift is expressed as a one sided alternative $H_a : \mathbb{E}[\phi_{\mathbf{Q}}] > \mathbb{E}[\phi_{\mathbf{P}}]$ (i.e., it is easier to learn how to reject on $\mathcal{Q}$ compared to $\mathcal{P}$). We refer to this test as *Detectron (Disagreement)*. To compute the test result at a significance level $\alpha$ we first estimate the null

distribution of $\phi_{\mathbf{P}}$ for a fixed sample size $n$ by training the Detectron for $K$ calibration rounds with different random seeds and sets $\mathbf{P}^\star$. The test result is significant if the observed disagreement rate $\phi_{\mathbf{Q}}$ is greater than the $(1-\alpha)$ quantile of the null distribution. For more information on the testing procedure see algorithm 1 below and a detailed description in Appendix B.

We consider an additional variant, *Detectron (Entropy)*, which computes the prediction entropy of each sample under the CDC ensemble instead of relying solely on disagreement rates. The CDC entropy is computed from the mean probabilities over each $N$ classes of the base classifier $f$ and set of $k$ CDCs $g_1, \ldots, g_k$.

$$\text{CDC}_{\text{entropy}}(x) = -\sum_{c=1}^{N} \hat{p}_c \log(\hat{p}_c) \ \text{ where } \ \hat{p} := \frac{1}{k+1}\left(f(x) + \sum_{i=1}^{k} g_i(x)\right) \qquad (4)$$

We use a KS test to compute a $p$-value for covariate shift directly on the entropy distributions computed for $\mathbf{Q}$ and $\mathbf{P}^\star$ and guarantee significance using the same strategy as above. The intuition for *Detectron (Entropy)* draws from the fact that when CDCs satisfy their objective (i.e., in the case of harmful shift) they learn to predict with high entropy on $\mathbf{Q}$ and low entropy on $\mathbf{P}^\star$, resulting in a natural way to distinguish between distributions.

---

**Algorithm 1:** The Detectron algorithm for detecting harmful covariate shift

---

**Input:** $\mathbf{P}$: labeled dataset, $\mathbf{Q}$: unlabeled dataset, $L$: learning algorithm,
$K$: calibration rounds $= 100$, $\aleph$: ensemble size $= 5$, $\alpha$: significance level $= 0.05$
**Output:** test result for harmful covariate shift at significance level $\alpha$

$\mathbf{P}_{\text{train}}$, $\mathbf{P}_{\text{val}}$, $\mathbf{P}^\star \leftarrow \text{Partition}(\mathbf{P})$
$N \leftarrow |\mathbf{Q}|$; $\Phi_{\mathbf{P}} \leftarrow [\,]$
$f \leftarrow L(\mathbf{P}_{\text{train}}, \mathbf{P}_{\text{val}})$            `// Load or train a base classifier on P`
**repeat**
    $\mathbf{p}^\star \leftarrow \text{RandomSample}(\mathbf{P}^\star, N)$
    `// Train an ensemble of CDCs on P`$^\star$
    **while** $n > 0$ *and iterations* $\leq \aleph$ **do**
        $g \leftarrow \text{ConstrainedDisagreement}(L, \mathbf{P}_{\text{train}}, \mathbf{P}_{\text{val}}, \mathbf{p}^\star, f)$    `// See Appendix TODO`
        $\mathbf{p}^\star \leftarrow \{x \mid x \in \mathbf{p}^\star \text{ and } f(x) = g(x)\}$    `// Filter out disagreed on data`
        $\phi_{\mathbf{P}} \leftarrow 1 - |\mathbf{p}^\star|/N$               `// Update disagreement rate`
    **end**
    Append $\phi_{\mathbf{P}}$ to $\Phi_{\mathbf{P}}$
**until** *K iterations elapse*
`// Train an ensemble of CDCs on Q`
**while** $n > 0$ *and iterations* $\leq \aleph$ **do**
    $g \leftarrow \text{ConstrainedDisagreement}(L, \mathbf{P}_{\text{train}}, \mathbf{P}_{\text{val}}, \mathbf{Q}, f)$
    $\mathbf{Q} \leftarrow \{x \mid x \in \mathbf{Q} \text{ and } f(x) = g(x)\}$
    $\phi_{\mathbf{Q}} \leftarrow 1 - |\mathbf{Q}|/N$
**end**
**return** $\phi_{\mathbf{Q}} > [(1-\alpha) \text{ *quantile of* } \Phi_{\mathbf{P}}]$

---

## 4 Empirical Evaluation

Our experiments are carried out on natural distribution shifts across multiple domains, modalities, and model types. We use the *CIFAR-10.1* dataset [Recht et al., 2019] where shift comes from subtle changes in the dataset creation processes, the *Camelyon17 dataset* [Veeling et al., 2018] for metastases detection in histopathological slides from multiple hospitals, as well as the *UCI heart disease* dataset [Janosi et al., 1988] which contains tabular features collected across international health systems and indicators of heart disease. We present unseen source an target domain performance of base models trained on source data in Appendix Table 4 which shows significant performance drops as an indicator for the hamrfulness of these shifts. See Appendix F for more details on datasets.

**Learning Constrained Disagreement.** We begin by training ensembles of 10 CDCs using the *disagreement cross entropy* (DCE) with CIFAR-10 as $\mathbf{P}$ and CIFAR-10.1 as $\mathbf{Q}$ for 100 random runs at a sample size of 50 (see Appendix E for training details). The results in Figure 2 empirically validates

minimizing the DCE as a CDC learning objective. The first observation is that when an unseen set is drawn from a shifted distribution $\mathcal{Q}$, the empirical disagreement rate $\phi_\mathbf{Q}$ grows significantly larger than the baseline disagreement rate $\phi_\mathbf{P}$. Next, we see that CDCs preserve accuracy on data from the training distribution. Finally, as the ensemble size increases (and disagreed upon points are removed) we see that the accuracy of the classifier increases. This indicates that the points that are disagreed upon early on in the algorithm are those that would have been misclassified.

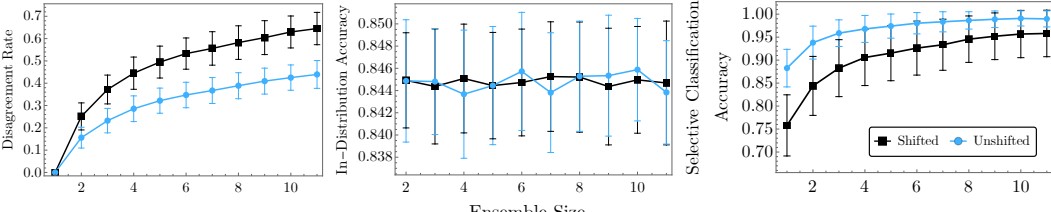

Figure 2: **Ensemble Size vs Properties of Constrained disagreement classifiers on CIFAR-10/10.1:** (Left) We see that for all ensemble sizes, there is lower disagreement on unshifted data (CIFAR-10) compared to disagreement on shifted data (CIFAR-10.1). (Center) Constrained disagreement does not compromise in-distribution performance. (Right) As the ensemble grows the selective classification accuracy computed on the set of test examples that all models agree on increases both on in-distribution and out-of-distribution data. Confidence intervals are computed as $\pm$ one standard deviation across experiments.

**Shift Detection Setup.** We evaluate the Detectron in a standard two-sample testing scenario similar to prior work [Zhao et al., 2022]. Given two datasets $\mathbf{P} = \{(x_i, y_i)\}_{i=1}^n$ ($x_i$ drawn from $\mathcal{P}$) and $\mathbf{Q} = \{\tilde{x}_i\}_{i=1}^m$ ($\tilde{x}_i$ drawn from $\mathcal{Q}$) and classifier $f$, we seek to rule out the null hypothesis ($\mathcal{P} = \mathcal{Q}$) at the $5\%$ significance level. To guarantee fixed significance we employ a permutation test by first sampling from the distribution of test statistics derived by the Detectron where the null hypothesis $\mathcal{P} = \mathcal{Q}$ holds (i.e., $\mathbf{Q}$ is drawn $\mathcal{P}$). We then compute a threshold over the empirical test statistic distribution that sets the false positive rate to $5\%$ (see Appendix B Figure 5). This step can be performed in advance of deployment as it only requires access to $\mathbf{P}$. To mimic deployment settings where we wish to identify covariate shift quickly, we assume access to far more samples from $\mathcal{P}$ compared to $\mathcal{Q}$. For each dataset, we begin by training a base classifier on the unshifted dataset. We evaluate the detection of covariate shift on 100 randomly selected test sets of $n = 10$, 20 and 50 samples from $\mathcal{Q}$. In all cases we train a maximum ensemble size of 5 (parameter $\aleph$ in algorithm 1). To prevent CDCs from overfitting in the case of small test set sizes, we perform early stopping if in-distribution validation performance drops by over 5% from the measured performance of the base classifier. Hyperparameters and training details for all models can be found in Appendix E.

**Evaluation.** We report the *True Positive Rate at 5% Significance Level (TPR@5)* aggregated over 100 randomly selected sets $\mathbf{Q}$. This signifies how often our method correctly identifies covariate shift ($\mathcal{P} \neq \mathcal{Q}$) while only incorrectly identifying shift 5% of the time. This is also referred to as the statistical power of a test at a significance level ($\alpha$) of 5%.

**Baselines.** We compare the Detectron against several methods for OOD detection, uncertainty estimation and covariate shift detection. *Deep Ensembles* Ovadia et al. [2019] using both (1) *disagreement* and (2) *entropy* scoring methods as a direct ablation to the CDC approach (3) *Black Box Shift Detection (BBSD)* [Lipton et al., 2018]. (4) *Relative Mahalanobis Distance (RMD)* [Ren et al., 2021]. (5) *Classifier Two Sample Test (CTST)* [Lopez-Paz and Oquab, 2017]. (6) *Deep Kernel MMD (MMD-D)* [Liu et al., 2020]. (7) *H Divergence (H-Div)* [Zhao et al., 2022]. For more information on baselines see Appendix subsection E.3.

**Shift Detection Experiments.** We begin with an analysis of the performance of the Detectron on the UCI Heart Disease dataset. Using a sample size ranging from 10 to 100 we compute the TPR@5 (based on 100 random seeds) and plot the results in Figure 3. We use gradient boosted trees (XGBoost [Chen and Guestrin, 2016]) for Detectron and CTST methods while the remaining baselines use a 2 layer MLP that achieves similar test performance. We report the mean and standard error of TPR@5 for sample sizes of 10, 20 and 50 on all datasets in Table 1.

### 4.1 DISCUSSION

*Overall Performance*. We observe in the bottom rows of Table 1 that Detectron methods outperform all baselines across all three tasks. This confirms our intuition that designing distribution tests

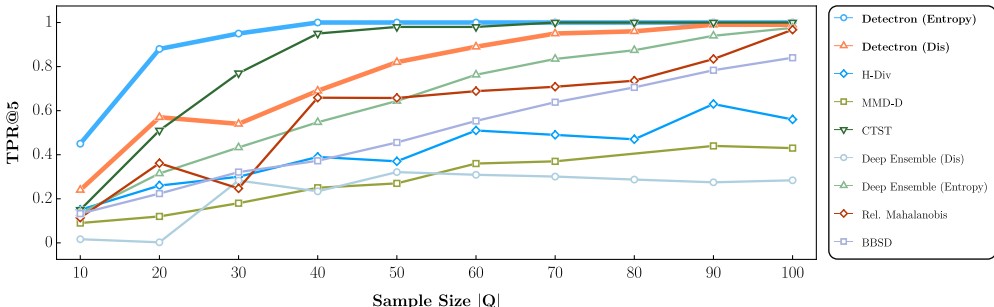

Figure 3: **True positive rate at the 5% significance level** for the Detectron and baseline methods for detection of covariate shift on the UCI heart disease dataset. The Detectron (Entropy) is shown to uniformly outperform baselines. Confidence intervals are excluded for visual clarity but are found in Table 1.

Table 1: **Results (true positive rate at the 5% significance level) for detection of harmful covariate shift** on CIFAR-10.1, Camelyon 17 and UCI Heart Disease benchmarks. The **best** result for each column is bolded, results that are within 2% of the best are underlined and the *best baseline* method is italicized.

| | CIFAR 10.1 | | | Camelyon 17 | | | UCI Heart Disease | | |
|---|---|---|---|---|---|---|---|---|---|
| **\|Q\|** | 10 | 20 | 50 | 10 | 20 | 50 | 10 | 20 | 50 |
| BBSD | $.07 \pm .03$ | $.05 \pm .02$ | $.12 \pm .03$ | $.16 \pm .04$ | $.38 \pm .05$ | $.87 \pm .03$ | $.13 \pm .03$ | $.22 \pm .04$ | $.46 \pm .05$ |
| Rel. Mahalanobis | $.05 \pm .02$ | $.03 \pm .03$ | $.04 \pm .02$ | $.16 \pm .04$ | $.40 \pm .05$ | *$.89 \pm .03$* | $.11 \pm .03$ | $.36 \pm .05$ | $.66 \pm .05$ |
| Deep Ensemble (Dis) | $.23 \pm .04$ | $.40 \pm .05$ | *$.74 \pm .04$* | $.10 \pm .03$ | $.11 \pm .03$ | $.13 \pm .03$ | $.02 \pm .01$ | $.00 \pm .00$ | $.32 \pm .05$ |
| Deep Ensemble (Entropy) | *$.33 \pm .05$* | *$.52 \pm .05$* | $.68 \pm .05$ | $.14 \pm .03$ | $.26 \pm .04$ | $.82 \pm .04$ | $.13 \pm .03$ | $.32 \pm .05$ | $.64 \pm .05$ |
| CTST | $.03 \pm .02$ | $.04 \pm .02$ | $.04 \pm .02$ | $.11 \pm .03$ | $.59 \pm .05$ | $.59 \pm .05$ | *$.15 \pm .04$* | *$.51 \pm .05$* | *$\underline{.98 \pm .01}$* |
| MMD-D | $.24 \pm .04$ | $.10 \pm .03$ | $.05 \pm .02$ | *$.42 \pm .05$* | *$.62 \pm .05$* | $.69 \pm .05$ | $.09 \pm .03$ | $.12 \pm .03$ | $.27 \pm .04$ |
| H-Div | $.02 \pm .01$ | $.05 \pm .02$ | $.04 \pm .02$ | $.03 \pm .02$ | $.07 \pm .03$ | $.23 \pm .04$ | *$.15 \pm .04$* | $.26 \pm .04$ | $.37 \pm .05$ |
| **Detectron (Dis)** | $\mathbf{.37 \pm .05}$ | $\underline{.54 \pm .05}$ | $.83 \pm .04$ | $.97 \pm .02$ | $\mathbf{1.0 \pm .00}$ | $.96 \pm .02$ | $.24 \pm 0.04$ | $.57 \pm 0.05$ | $.82 \pm 0.04$ |
| **Detectron (Entropy)** | $\underline{.35 \pm .05}$ | $\mathbf{.56 \pm .05}$ | $\mathbf{.92 \pm .03}$ | $\mathbf{.97 \pm .02}$ | $\mathbf{1.0 \pm .00}$ | $\mathbf{1.0 \pm .00}$ | $\mathbf{.45 \pm .05}$ | $\mathbf{.88 \pm 0.03}$ | $\mathbf{1.0 \pm .00}$ |

based specifically on available data and the outputs of learning algorithms is a promising avenue for improving existing methods in the high dimensional/low data regime.

*Sample Efficiency.* For more significant shifts (Camelyon and UCI), we see in Table 1 the most significant improvements over baselines in the lowest sample regime (10 data points). The fine-grained result in Figure 3 shows that CTST catches up to Detectron at 40 samples while deep ensemble, BBSD, and Mahalanobis catch up at 100.

*Disagreement vs Entropy.* For the experiments on imaging datasets with deep neural networks Detectron (Disagreement) often performs nearly as well as Detectron (Entropy), while Detectron (Entropy) is strictly superior for the UCI dataset. While we recommend entropy as the method to maximize test power, disagreement is a more interpretable statistic as it is correlates well with the portion of misclassified samples (see (right) Figure 2).

*Comparison to baselines.* Amongst the baselines, there is no clear best method. On average, ensemble entropy is superior on CIFAR, MMD-D on Camelyon, and CTST on UCI. Our method may be thought of as a combination of ensembles, CTST, and H-Divergence. As ensembles, we leverage the variation in outputs between a set of classifiers; as CTST, we learn in a domain adversarial setting; and as H-Divergence, we compute a test statistic based on data that a model was trained on. Lastly, while MMD-D and H-Divergence were shown to be the previous state-of-the-art, their performance was validated only on larger sample sizes ($\geq 200$).

*On Tabular Data.* The Detectron shows promise for deployment on tabular datasets (bottom right of Table 1 and Figure 3), where (1) the computational cost of training models is low, (2) the model agnostic nature of the Detectron is beneficial as random forests often outperform neural networks in tabular data [Borisov et al., 2021], and (3) based on our discussions with medical professionals, the ability to detect covariate shift from small test sizes is of particular interest in the healthcare domain where population shift is a constant problem burden for maintaining the reliability of deployed models.

*On computational cost:* Our method is more computationally expensive than some existing methods for detecting shifts such as BBSD and Mahalanobis Scores, but is similar complexity to other approaches such as Ensembles, MMD-D and H-Divergence which may require training multiple deep models. However, as the Detectron leverages a pretrained model already in deployment, we find in practice that only a small number of training rounds are required to create each CDC. Furthermore, looking at the runtime behavior in Figure 4 we see that while allowing for more computation time increases the fidelity of the Detectron, only a small number of training batches may be required to achieve a desirable level of statistical significance. In scenarios where the deployed classifier is deemed high-risk (e.g. healthcare, justice system, education) we believe the additional computational expense is justified for an accurate assessment of whether the classifier needs updating. Having established the utility, accelerating the Detectron as well as building a deeper understanding of the runtime performance is fertile ground for future work.

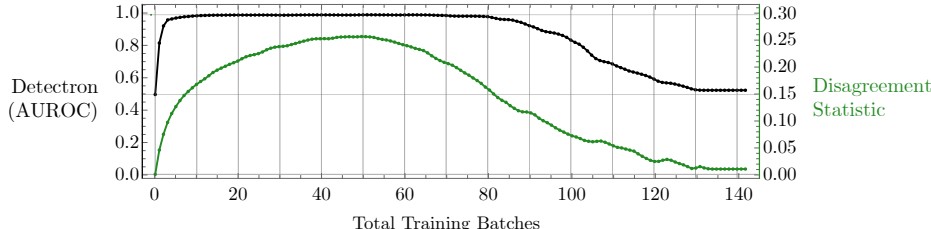

Figure 4: **Runtime Characteristics:** We train 100 random runs of CDCs on 100 samples from CIFAR 10 and 10.1 and compute the *disagreement satistic* as the difference $\psi := \mathbb{E}[\phi_{\mathbf{Q}} - \phi_{\mathbf{P}}]$. While we see that while $\psi$ peaks near 50 training batches, only 10 batches are required for the Detectron disagreement test to reach an area under the TPR vs FPR curve (AUROC) of nearly 1 (i.e., perfect discrimination). Training CDCs for too long eventually lowers $\psi$ as $\mathbb{E}[\phi_{\mathbf{Q}}] \approx \mathbb{E}[\phi_{\mathbf{P}}] \approx 1$ meaning CDCs eventually overfit to disagreeing on all of their data.

## 5  CONCLUSION, LIMITATIONS AND FUTURE WORK

Our work presents a practical method capable of detecting covariate shifts given a pre-existing classifier. On both neural networks and random forests, we showcase the efficacy of our method to detect covariate shift using a small number of unlabelled examples across several real-world datasets. We remark on several characteristics that represent potential directions for future work:

*On Generalization and Model Complexity:* The definition of the Detectron specifies the use of the same function class for identifying shift as is used in the original prediction problem. The Detectron may fail to detect harmful shifts in cases where the base model is learned from an underspecified function class. Appendix I provides additional context, examples and ways to mitigate this limitation. The precise relationships between model complexity, generalization error, and test power are interesting directions for future work.

*Beyond Classification:* Our work here focuses on the case of classification, however, we believe there is a viable extension of our work to regression models where constrained predictors are explicitly learned to maximize test error according to the existing metric, such as mean squared error. We leave this exploration for future work.

Finally, we wish to highlight that while auditing systems such as the Detectron show promise to ease concerns when using learning systems in high-risk domains, practitioners interfacing with these systems should not place blind trust in their outputs.

## ETHICS STATEMENT

The speed of the adoption of ML into risk scenarios raises a critical need for methods that ensure trustworthiness and reliability. However, over-reliance on such methods brings about critical ethical considerations. As we have seen, Detectron is highly sensitive to picking out discriminative features of data distributions, and, as such, its usage may prevent practitioners from deploying models in new environments. As a result, the individuals in those environments may become subject to unfair treatment. For instance, if Detectron determines a model trained on hospital A safe to deploy in hospitals B and C, but not in D, the individuals in population D may experience a lower level of care. In a real example, Detectron, when tested on a model trained on a subset of light-skinned celebrities (CelebA dataset–Liu et al. [2015]), quickly raises the alarm when given images of those that are not light-skinned. While Detectron can help mitigate potential disasters encountered by deploying models in hazardous domains, it should not be an excuse for practitioners to avoid collecting richer and more diverse datasets as a primary strategy to ensure model reliability.

## ACKNOWLEDGMENTS

Tom Ginsberg's research was supported by a New Frontiers in Research Fund NFRFR-2022-00526, an LG research grant, and a Canada Graduate Scholarship (CGS-M). Rahul G. Krishnan was supported by a CIFAR AI Chair. Resources used in preparing this research were provided, in part, by the Province of Ontario, the Government of Canada through CIFAR, and companies sponsoring the Vector Institute. Additional thanks is given to the many readers who provided valuable feedback: Vahid Balazadeh, Michael Cooper, Edward De Brouwer, Aslesha Pokhrel, Adnan Mohd, Ian Shi, Asic Chen and Stephan Rabanser.

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

# Appendix

## Table of Contents

# A    LEARNING ALGORITHMS

This section presents further details on the primary learning algorithms used and referred to in our work.

## A.1    REJECTRON

We provide a summary of the original Rejectron algorithm for PQ learning [Goldwasser et al., 2020] as it is the primary motivation for our work. Rejectron (algorithm 2) takes as input a labeled training set of $n$ samples $\mathbf{x}$ (iid over $\mathcal{P}$), an unlabeled test set of $n$ samples $\tilde{\mathbf{x}}$ (iid over $\mathcal{Q}$), an error $\epsilon$ and a weight $\Lambda$. The output is a selective classifier, that predicts according to a base classifier $h$ if the input $x$ is inside some set $S$ and otherwise rejects (abstains from predicting).

$$h|_S(x) := \begin{cases} h(x) & x \in S \\ \text{reject} & x \notin S \end{cases} \tag{5}$$

The error and rejection rate of a selective classifier are defined for a selective classifier $h|_S$ with respect to a distribution $\mathcal{P}$ as:

$$\text{rej}_h(S) := \mathbb{P}_{x \sim \mathcal{P}}[x \notin S] \quad \text{err}_h(S) := \mathbb{P}_{x \sim \mathcal{P}}[h(x) \neq y \wedge x \in S] \tag{6}$$

The *empirical* rejection and error rates for a set of samples $\mathbf{x} = \{x_i\}_{i=1}^n$ and corresponding labels $\mathbf{y} = \{y_i\}_{i=1}^n$ are similarly defined as:

$$\text{rej}_h(\mathbf{x}) := \frac{1}{n}\sum_{i=1}^n \mathbb{1}_{x_i \notin S} \quad \text{err}_h(\mathbf{x}) := \frac{1}{\sum_{i=1}^n x_i \in S}\sum_{i=1}^n \left(\mathbb{1}_{x_i \neq y_i} \times \mathbb{1}_{x_i \in S}\right) \tag{7}$$

Under the selective classification framework Goldwasser et al. extend the conventional concept of PAC learning [Haussler, 1990] to test samples drawn from an unknown distribution $\mathcal{Q}$.

*PQ learning [Goldwasser et al., 2020]* Learner $L$ $(\epsilon, \delta, n)$-PQ-learns $F$ for $0 \leq \epsilon \leq 1, 0 \leq \delta \leq 1$ and $n \in \mathbb{Z}^+$ if for any distributions $\mathcal{P}, \mathcal{Q}$ over $X$ and any ground truth function $d \in F$, its output $h := L(\mathbf{P}, d(\mathbf{P}), \mathbf{Q})$ satisfies

$$\mathbb{P}_{\mathbf{x} \sim \mathcal{P}^n, \tilde{\mathbf{x}} \sim \mathcal{Q}^n}\left[\text{rej}_h(\mathbf{x}) + \text{err}_h(\tilde{\mathbf{x}}) \leq \epsilon\right] \geq 1 - \delta \tag{8}$$

Under several assumptions and a special value $\epsilon^\star$, this selective classifier is guaranteed with high probability to have error less then $2\epsilon^\star$ on $\tilde{\mathbf{x}}$ and a rejection rate below $\epsilon^\star$ on $\mathbf{x}$ (see Theorem 5.7 in Goldwasser et al.).

---

**Algorithm 2:** Rejectron [Goldwasser et al., 2020]

---

**Input:** train $\mathbf{x} \in X^n$, labels $\mathbf{y} \in Y^n$, test $\tilde{\mathbf{x}} \in X^n$, error $\epsilon \in [0, 1]$, weight $\Lambda = n + 1$
**Output:** selective classifier $h|_S$
$h \leftarrow \text{ERM}(\mathbf{x}, \mathbf{y})$
**for** $t = 1, 2, 3, \ldots$ **do**
 1. $S_t := \{x \in X : h(x) = c_1(x) = \ldots = c_{t-1}(x)\}$
 2. Choose $c_t \in C$ to maximize $s_t(c) := \text{err}_{\tilde{\mathbf{x}}}\left(h|_{S_t}, c\right) - \Lambda \cdot \text{err}_{\mathbf{x}}(h, c)$ over $c \in C$
 3. If $s_t(c_t) \leq \epsilon$, then stop and return $h|_{S_t}$

---

Rejectron starts by querying an empirical risk minimization (ERM) oracle that uses a 0–1 risk score over a concept class $C$ for a model $h$ that perfectly learns the training dataset. A primary assumption for Rejectron to output a perfect model as well as for it to eventually find a selective classifier that meets the $\epsilon^\star$ bound is that the true decision function (i.e., the function that creates the training labels) is also a member of $C$. The authors refer to this setting as *realizable*.

On the first iteration of the algorithm, Rejectron finds another model $c_1 \in C$ that jointly maximizes the error with respect to $h$ on $\tilde{\mathbf{x}}$ while minimizing it on $\mathbf{x}$. The authors show that they can efficiently solve this optimization problem using a single ERM query on a dataset of $n^2 + n$ samples (see Lemma 5.1 in Goldwasser et al.).

In every subsequent step $t > 1$ a set $S_t$ is created where all models $h$ through to $c_{t-1}$ agree. Another model $c_t \in C$ is found that maximizes the same objective as above but only on the intersection of $\tilde{\mathbf{x}}$ and $S_t$. Upon termination a selective classifier $h|_{S_t}$ is output.

## A.2 CONSTRAINED DISAGREEMENT CLASSIFIERS

We formally present the algorithm for creating a constrained disagreement classifier (section 3), a fundamental tool used to detect distribution shift in our work. The main inputs are a labeled training set, an unlabeled test set, and a classifier trained on $\mathbf{P}$ using a learning algorithm $L$. Three other hyperparameters include a metric to evaluate the performance of a classifier on a labeled dataset (e.g., accuracy), a tolerance $\epsilon$ which controls how much the metric may drop during disagreement steps and and a maximum number of epochs.

---

**Algorithm 3:** Constrained Disagreement

---

**Input:** $L$: learning algorithm, $\mathbf{P}_{\text{train}}$: labeled training dataset $\{\ldots, (x_i, y_i), \ldots\}$, $\mathbf{P}_{\text{val}}$: labeled validation dataset $\{\ldots, (x_i, y_i), \ldots\}$, $\mathbf{Q}$: unlabeled test dataset $\{\ldots, x_i, \ldots\}$, $f$: classifier trained on $\mathbf{P}$, $\mathcal{M}$: evaluation metric (default *accuracy*), $\varepsilon$: tolerance (default 0.05), $k$: max epochs (default 10)

**Output:** Constrained Disagreement Classifier $g_{(f, \mathbf{P}, \mathbf{Q})}$

$\hat{\mathbf{Q}} \leftarrow \{(x, f(x)) \mid \hat{x} \in \mathbf{Q}\}$          // infer pseudo labels on $\mathbf{Q}$ using $f$
                        // create a batched dataloader using $\mathbf{P}$ and $\mathbf{Q}$
$\mathbf{PQ} \leftarrow \text{Batched}(\{(x, y) \mid (x, y) \in \mathbf{P} \wedge (x, y) \in \mathbf{Q}\})$ $g \leftarrow f$    // Initialize $g$ with $f$
$m_0 \leftarrow \mathcal{M}(f, \mathbf{P}_{\text{val}})$            // Compute the validation performance of $f$
**while** $\mathcal{M}(f, \mathbf{P}_{val}) > m_0 - \varepsilon$ *and iterations* $< k$ **do**
                           // Training epoch over $\hat{\mathbf{PQ}}$
    **for** *batch* $\in \mathbf{PQ}$ **do**
        $x_P, y_P \leftarrow \{(x, y) \mid (x, y) \in \text{ batch and } (x, y) \in \mathbf{P}_{\text{train}}\}$
        $x_Q, y_Q \leftarrow \{(x, y) \mid (x, y) \in \text{ batch and } (x, y) \in \hat{\mathbf{Q}}\}$
        // Update $g$ using an existing learning algorithm for $(x_P, y_P)$ and the appropriate disagreement update for $(x_Q, y_Q)$
        $g \leftarrow \text{Update}(g, L, (x_P, y_P), (x_Q, y_Q))$
    **end**
**end**
**return** $g$

---

Our algorithm is a practical generalization of the inner step in Rejectron:
(1) We require tight performance monitoring on a validation set to prevent overfitting as we commonly observed that training models to disagree on small in-distribution test sets cause a drop in their in-distribution performance on unseen data after many training epochs. (2) We allow for arbitrary-sized train and test sets. (3) We provide a methodology (section 3 *Learning to Disagree with the Disagreement Cross Entropy*) for updating arbitrary classification models to disagree on test data while agreeing on in-distribution data that leverages the generalization structure of their original learning algorithm while requiring quadractically fewer training samples then Rejectron. Further details on implementation details for step (3) can be found below in Appendix D.

## B HYPOTHESIS TESTS

### B.1 TWO SAMPLE TESTING METHODOLOGY

Our method to detect covariate shift, like prior work, is to perform a statistical hypothesis test between the distributions of one or low dimensional quantities derived using each element of a possibly shifted target dataset $\mathbf{Q}$ and a known in-distribution source dataset $\mathbf{P}^\star$ that has not been observed in during model development. A significant motivation for our work was that the majority of statistical hypothesis tests used by prior work are formulated in a fashion that is independent of the particular target dataset being tested (e.g., BBSD [Lipton et al., 2018] which uses a pre-trained classifier as an ansatz for a dimensionality reduction). However, with the Detectron we follow a transductive approach by building a statistical test by training classifiers to meet a carefully crafted objective (i.e., constrained disagreement) on the target data. A drawback of this approach is that low dimensional representations are, in general, not be iid; hence to perform a fair statistical test, we must run the Detectron on a known in-distribution dataset under the **same** experimental conditions (e.g., sample size, learning rate, ensemble size). For other baseline methods that do not take the

transductive approach (e.g., Mahalanobis, BBSD, Ensemble), we are not limited to choosing a source dataset $\mathbf{P}^\star$ of the same size as the samples can again be assumed to be iid. In practice, for iid methods, we fix the size of $\mathbf{P}^\star$ to 1000 for CIFAR and Camelyon, and in UCI Heart Disease, we use only 120 as the dataset is significantly smaller (920 samples).

## B.2 Statistical Tests Used in Methods/Baselines

We provide a summary in the context of our work on the three types of statistical tests used in our experiments. We also explain technical details on how we use each test in our experiments.

**Kolmogorov–Smirnov (KS) Test.** The KS test is one of the most common non-parametric univariate statistical tests. It the two sample setting given datasets $X = \{x_1, \ldots, x_n\}$ and $Y = \{y_1, \ldots, y_m\}$ the test statistic is computed as the maximum difference between the empirical CDFs of $X$ and $Y$. An asymptotically correct $p$-value can is computed using a closed-form expression of the test statistic and sample sizes $n$ and $m$. An exact $p$-value can also be found by considering the fraction of every possible pair of empirical CDFs that lie within the region with a maximum bounded difference; more details can be found in Hodges [1958]. In practice we use the KS test implementation found in `scipy.stats.ks_2samp` which automatically computes exact $p$-values when $\max\{n, m\} \leq 10,000$ and otherwise defaults to the asymptotic approximation.

We use KS tests for any distributions derived from continuous scores within our methods and baselines. For the *Relative Mahalanobis Score* test [Ren et al., 2021], we compute the $p$-value for shift via a KS test between the Mahalanobis score for the possibly shifted target data $\mathbf{Q}$ and a source dataset of unseen in distribution samples $\mathbf{P}^\star$. In *BBSD* [Lipton et al., 2018] we compute a KS test on each dimension of the softmax output of a classifier between the source and target datasets, the final $p$-value is found via Bonferroni correction as is done by Rabanser et al. [2019], which simply takes the minimum $p$-value and divides it by the number of tests (e.g the softmax dimension). Similarly, using *Deep Ensemble (Entropy)* and *Detectron (Entropy)*, we perform a KS test directly on the distribution of entropy values computed from each sample in the source and target datasets, respectively. See Figure 6 for a full description of the Detectron entropy test.

**Binomial Test.** The binomial test is simple to state and has an elegant closed-form solution. We consider a binomially distributed random variable with rate $q$ $X \sim \text{Binomial}(n, q)$ for which we observe a single sample $x$. Since the binomial distribution is defined as a sum of iid Bernoulli random variables with the same rate, $x$ may equivalently be interpreted as a set of $n$ samples of which $x$ are 1 and $n - x$ are 0. Given a a baseline rate $p$ we wish to determine the probability of observing an event at least as rare as $X = x$ under the null hypothesis that $p = q$, this quantity can be computed exactly using the symmetry of the binomial distribution.

$$\mathbb{P}_{X \sim \text{Bin}(n,p)}(X \text{ is rarer then } x) = 2 \times \mathbb{P}_{X \sim \text{Bin}(n,p)}(X \geq x)$$

$$= 2 \sum_{k=x}^{n} \mathbb{P}[X = k]$$

$$= 2 \sum_{k=x}^{n} (1-p)^{n-k} p^k \binom{n}{k} = 2 \frac{B_p(x, n-x+1)}{B(x, n-x+1)}$$

Where $B_z(\alpha, \beta)$ is the incomplete Beta function and $B(\alpha, \beta)$ is the beta function. Binomial testing is used in the *Deep Ensemble (Disagreement)* baseline method where we estimate $p$ as the disagreement rate of a deep ensemble on the set $\mathbf{P}^\star$ (i.e the number of samples in $\mathbf{P}^\star$ where the ensemble does not predict unanimously divided by the size of $\mathbf{P}^\star$) and test for distribution shift based on the result of a binomial test on the observed disagreement on $\mathcal{Q}$. Binomial testing is also used for the classifier two sample test method (CTST) [Lopez-Paz and Oquab, 2017]. First a domain classifier is trained to separate source and target data then its performance is tested on a set of unseen data where the number of samples of a total of $N$ it correctly assigns a domain label to is compared to the null distribution $\text{Bin}(N, 0.5)$ (i.e. random guessing). For implementation purposes we use `scipy.stats.binomtest`.

**Permutation Test.** Our ultimate goal is to detect covariate shift $\mathcal{P} \neq \mathcal{Q}$ at a bounded significance level (i.e. bounded probability of outputting $\mathcal{P} \neq \mathcal{Q}$ when in fact $\mathcal{P} = \mathcal{Q}$). To bound the significance level, we follow the simple and principled approach of the permutation test. Suppose we wish to run the Detectron to test for shift on a set $\mathbf{Q}$ from a baseline $\mathbf{P}^\star$ (each of $N$ samples) while Detectron,

or any other test, computes a $p$-value on some low dimensional samples derived from $\mathbf{Q}$ and $\mathbf{P}^\star$, the significance threshold on that test will not in general correspond precisely to the significance for rejecting the original null hypothesis $\mathcal{P} = \mathcal{Q}$. The permutation test allows us to reclaim statistical guarantees by first performing several tests where the null hypothesis holds (e.g., we draw $\mathbf{Q}$ from $\mathcal{P}$) and find a cutoff for the significance of a $p$-value that sets the false positive rate at exactly $5\%$.

Our experiments run the Detectron 100 times for each sample size on random sets $\mathbf{Q}$ drawn from $\mathcal{P}$. Based on these runs we compute $95^{\text{th}}$ percentile ($\tau$) on the final rejection rate. We then run the actual test using a set of samples $\mathbf{Q}$ drawn from $\mathcal{Q}$ which we deem significant at the $5\%$ level if the number of rejected samples is greater than $\tau$. A visual description of this method can be found in $Figure\ 5$.

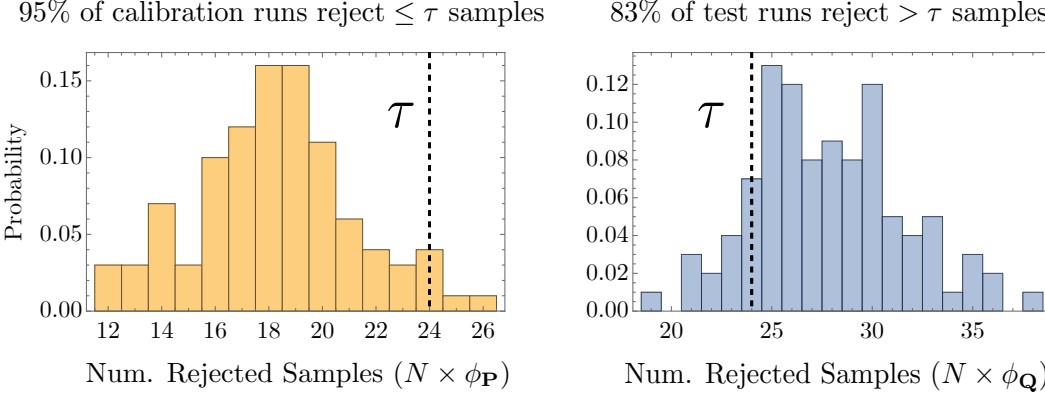

Figure 5: **The Detectron disagreement test:** In this example (taken from our experiment where $\mathcal{P}$ = CIFAR10 and $\mathcal{Q}$ = CIFAR10.1 and sample size $N = 50$) pictured we start by training an ensemble of CDCs (we use and ensemble size of 5) to reject/disagree on a set of $N$ unseen samples from the original training distribution ($\mathbf{P}^\star$) while constrained to perform consistently with a base model on the original training and validation sets used to train the base model on CIFAR10. We perform 100 of these calibration runs using different random seeds and samples for $\mathbf{P}^\star$ to estimate a threshold $\tau$ such that $95\%$ of the runs reject fewer then $\tau$ samples — thereby fixing the significance level of the test to $5\%$. To estimate the test power, we train CDCs using **the exact same configuration** as the calibration runs except we replace $\mathbf{P}^\star$ with a random set of $N$ samples $\mathbf{Q}$ from $\mathcal{Q}$ (CIFAR 10.1). By averaging the number of runs the reject more than $\tau$ samples we can compute the power (or true positive rate) of the test for the configuration.

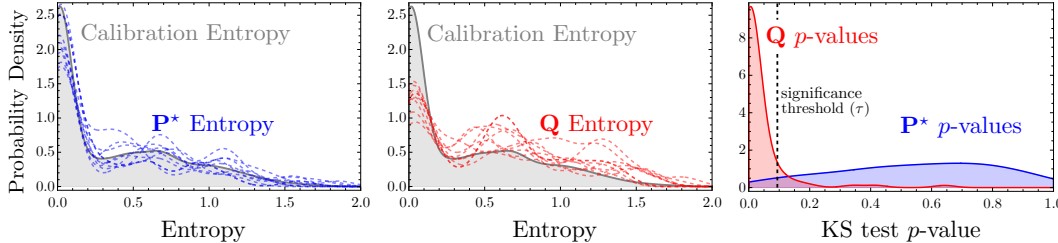

Figure 6: **The Detectron entropy test:** Following the same experimental setup as Figure 5, we start (left) by computing a KS test between the continuous entropy values for each calibration run $\mathbf{P}^\star$ with the flattened set of entropy values from all other 99 calibration runs. Then (center) we compute a KS test from each test run $\mathbf{Q}$ with a random set of all but one calibration runs. Finally (right), we find a threshold $\tau$ on the distribution of $p$-values obtained from step 1 as the $\alpha$ quantile to guarantee a false positive rate of $\alpha$. The power of the test is computed as the fraction of $p$-values computed from 100 test runs $\mathbf{Q}$ that are below $\tau$.

## C    PROOFS

The full proof of Theorem 1 referenced in section 3 *From Constrained Disagreement to Detecting Shift with Hypothesis Tests* is presented below. After which, we prove a related result that invokes a Bayesian perspective on distribution shift to generate a tight bound on the probability of shift given observations made by CDCs on unseen data.

**Theorem 1** (Disagreement implies covariate shift). *Let $f$ be a classifier trained on dataset $\mathbf{P}$ consisting of samples drawn identically from $\mathcal{P}$ and their corresponding labels. Let $g$ be a classifier that is observed to agree (classify identically) with $f$ on $\mathbf{P}$ but and disagree on a dataset $\mathbf{Q}$ drawn from $\mathcal{Q}$. If the rate which $g$ disagrees with $f$ on $n$ unseen samples from $\mathcal{Q}$ is greater then that from $n$ unseen samples from $\mathcal{P}$ w.p greater than $p^\star := \frac{1}{2}\left(1 - 4^{-n}\binom{2n}{n}\right)$ there must be covariate shift.*

*Proof.* Let $R_P = \emptyset_1^P + \ldots + \emptyset_n^P$ where each $\emptyset_i^P$ is an i.i.d Bernoulli random variable that describes the probability of $g$ disagreeing with $f$ on an unseen sample from $\mathcal{P}$. Additionally, let $R_Q = \emptyset_1^Q + \ldots + \emptyset_n^Q$ be defined similarly for $\mathcal{Q}$. If $\mathcal{P}$ and $\mathcal{Q}$ are equal then $\emptyset_i^P$ and $\emptyset_i^Q$ are equal by definition and the probability of observing $R_Q > R_P$ is tightly bounded by Equation 9. This is the tightest upper bound that is not a function of $\mathbb{E}[\emptyset_i^P]$, the proof can be found in Lemma 1.

$$\mathcal{P} = \mathcal{Q} \implies \mathbb{P}\left(R_Q > R_P\right) \leq \frac{1}{2}\left(1 - 4^{-n}\binom{2n}{n}\right) = \frac{1}{2} - O\left(\frac{1}{\sqrt{n}}\right) \tag{9}$$

The more helpful contrapositive statement says that if it is sufficiently likely that $R_Q > R_P$, then covariate distributions $\mathcal{P}$ and $\mathcal{Q}$ must not be equal.

$$\mathbb{P}\left(R_Q > R_P\right) > \frac{1}{2}\left(1 - 4^{-n}\binom{2n}{n}\right) \implies \mathcal{P} \neq \mathcal{Q} \tag{10}$$

This result naturally lends itself to identifying $\mathcal{P} \neq \mathcal{Q}$ by rejecting an exact statistical hypothesis that $R_Q = R_P$ in favor of the alternative $R_Q > R_P$. $\qquad\square$

**Lemma 1.** Let $X$ and $Y$ be iid binomial random variables with distribution $\text{Bin}(n,p)$ then for all $n \in \mathbb{Z} \geq 0$:

$$P(X > Y) \leq \frac{1}{2}\left(1 - 4^{-n}\binom{2n}{n}\right) < \frac{1}{2} \tag{11}$$

Furthermore, eq. (11) is the tightest possible bound that does not depend on $p$.

*Proof.* Let $Z$ be the distribution $X - Y$, while $Z$ itself is intractable to write down for arbitrary $n$, the characteristic function takes a convenient form

$$\phi_Z(t;p) = \mathbb{E}\left[e^{it(x-y)}\right] = \mathbb{E}\left[e^{itx}\right]\mathbb{E}\left[e^{-ity}\right] \tag{12}$$

$$= \left(1 + p\left(-1 + e^{it}\right)\right)^n \left(1 + p\left(-1 + e^{-it}\right)\right)^n \tag{13}$$

$$= \left(-p^2 e^{-it} - p^2 e^{it} + 2p^2 + pe^{-it} + pe^{it} - 2p + 1\right)^n \tag{14}$$

$$= (1 - 2p + p\cos(t) - ip\sin(t) + p\cos(t) + ip\sin(t) \tag{15}$$
$$+ 2p^2 - p^2\cos(t) + ip^2\sin(t) - p^2\cos(t) - ip^2\sin(t))^n$$

$$= \left(p^2(2 - 2\cos(t)) + p(2\cos(t) - 2) + 1\right)^n \tag{16}$$

Since $X$ and $Y$ are identically distributed $P(Z > 0) = P(Z < 0)$ and so

$$P(Z > 0) = \frac{1}{2}\left(1 - P(Z = 0)\right) \tag{17}$$

Equation (17) suggests that a tight upper bound of the form $P(Z = 0) \geq \alpha$ implies a tight lower bound in the form $P(Z > 0) \leq (1 - \alpha)/2$. To bound $P(Z = 0)$ we first write it as an integral expression using the characteristic inversion formula for discrete random variables [Ushakov, 2011]

$$P(Z = 0) = \frac{1}{2\pi}\int_{-\pi}^{\pi}\phi_Z(t;p)dt \tag{18}$$

Since $\phi_Z(t; p)$ has the form $(a(t)p^2 + b(t)p + 1)^n$ where $a$ and $b$ are real valued functions and $a(t) \geq 0 \ \forall t \in \mathbb{R}$ (i.e an integer power of a quadratic equation with positive leading coefficient), then for any choice of $t$, $\phi_Z(t; p)$ will be globally minimized if and only if $p \to p^\star = -b(t)/(2a(t))$. For the particular form of $\phi_Z(t; p)$, $p^\star$ is simply $1/2$

$$p^\star = -\frac{b(t)}{2a(t)} = -\frac{2\cos(t) - 2}{2(2 - 2\cos(t))} = \frac{1}{2} \tag{19}$$

This result is intuitive as the variance of a binomial distribution $\mathrm{Bin}(n, p)$ is maximized for any fixed choice of $n$ when $p = 1/2$. We should note that Equation 19 appears problematic when $\cos(t) = 1$, but in this case $\phi(t; p)$ becomes constant, hence $p$ cannot influence the upper bound. We can now write the upper bound for $P(Z = 0)$

$$P(Z = 0) = \frac{1}{2\pi} \int_{-\pi}^{\pi} \phi_Z(t; p) dt \tag{20}$$

$$\geq \frac{1}{2\pi} \int_{-\pi}^{\pi} \phi_Z\left(t; \frac{1}{2}\right) dt \tag{21}$$

$$= \frac{1}{2\pi} \int_{-\pi}^{\pi} 2^{-n}(\cos(t) + 1)^n dt \tag{22}$$

$$= 4^{-n} \binom{2n}{n} \tag{23}$$

The final expression in eq. (23) can be found using the change of variables $z = e^{it}$ and Cauchy's residue theorem [Needham, 2000]

$$I = \frac{1}{2\pi} \int_{-\pi}^{\pi} 2^{-n}(\cos(t) + 1)^n dt \tag{24}$$

$$= -\frac{i}{2\pi} \oint_{|z|=1} 2^{-n} \frac{1}{z}\left(1 + \frac{1 + z^2}{2z}\right)^n dz \ (\text{using } t \to -i\log z) \tag{25}$$

$$= -\frac{i}{2\pi} \oint_{|z|=1} 4^{-n} z^{-n-1}(z + 1)^{2n} dz \ (\text{simplifying}) \tag{26}$$

$$= \mathrm{Res}\left(4^{-n} z^{-n-1}(z + 1)^{2n}, 0\right) \ (\text{applying Cauchy's Theorem}) \tag{27}$$

$$= \frac{1}{4^n n!} \lim_{z \to 0} \left(\frac{d^n}{dz^n}(z + 1)^{2n}\right) \tag{28}$$

$$= \frac{1}{4^n n!} 2n(2n - 1)(2n - 2)\ldots(n + 1) \tag{29}$$

$$= 4^{-n} \binom{2n}{n} \tag{30}$$

Combining eq. (17) with the bound from eq. (23) we arrive at the conjectured upperbound

$$P(X > Y) = P(Z > 0) = \frac{1}{2}(1 - P(Z = 0)) \tag{31}$$

$$\leq \frac{1}{2}\left(1 - 4^{-n}\binom{2n}{n}\right) \ \text{by eq. (23)} \tag{32}$$

$$\tag{33}$$

Finally we may use Sterling's approximation to show that $4^{-n}\binom{2n}{n} \in O\left(\frac{1}{\sqrt{n}}\right)$ and hence converges to 0 as $n \to \infty$ leaving a limiting tight upper bound of $1/2$. $\qquad \square$

**Theorem 2** (Probability of Disagreement: A Bayesian Perspective). Let $f$ be a classifier trained on dataset $\mathbf{P}$ drawn from the distribution $\mathcal{P}$ over $X$ and their corresponding labels. Let $g$ be a classifier observed to agree with $f$ on $\mathbf{P}$ but disagree on a dataset $\mathbf{Q}$ drawn from a distribution $\mathcal{Q}$ over $X$. We denote the true probabilities that $g$ will disagree with $f$ on a sample from $\mathcal{P}$ and $\mathcal{Q}$ as $p$ and $q$, respectively. Under a uniform prior $\mathcal{U}(0, 1)$ for $p$ and $q$, if we observe that $g$ disagrees with $f$ on

$m$ out of $M$ iid samples from $\mathcal{Q}$ while disagreeing with $n$ out of $N$ iid samples from $\mathcal{P}$, then the posterior probability that $g$ is truly more likely to disagree with $f$ on $\mathcal{Q}$ compared to $\mathcal{P}$:

$$\mathbb{P}[q > p] = 1 - \frac{(M+1)!(N+1)!(m+n+1)!}{(m+1)!n!(M-m)!(m+N+2)!} \times \tag{34}$$
$${}_3F_2(m+1, m-M, m+n+2; m+2, m+N+3; 1)$$

Where ${}_pF_q$ is the generalized hypergeometric function, implemented in several standard mathematical libraries.

*Proof.* For simplicity we consider the function dis : $X \to \{0, 1\}$ that outputs 0 if $f(x) = g(x)$ else 1. We define the true disagreement rates **p** and **q** as

$$\mathbf{p} := \mathbb{E}_{x \sim \mathcal{P}}[\text{dis}(x)] \text{ and } \mathbf{q} := \mathbb{E}_{x \sim \mathcal{Q}}[\text{dis}(x)] \tag{35}$$

Without any *a-priori* knowledge of dis we define the random variables $p$ and $q$ under uniform prior (i.e $p, q \overset{\text{i.i.d}}{\sim} \mathcal{U}(0, 1)$) to encode our belief over the true values **p** and **q**. Now we draw $N$ samples as $\mathbf{x} \overset{\text{i.i.d}}{\sim} \mathcal{P}^N$ and $M$ as $\tilde{\mathbf{x}} \overset{\text{i.i.d}}{\sim} \mathcal{Q}^M$ and compute the number of times dis$(x)$ equals 1 on each set.

$$n := \sum_{x \in \mathbf{x}} \text{dis}(x) \text{ and } m := \sum_{x \in \tilde{\mathbf{x}}} \text{dis}(x) \tag{36}$$

By definition in Equation 35 we know that $n$ and $m$ are draws from binomial distributions: $\mathcal{N} \sim \text{Bin}(N, p)$ and $\mathcal{M} \sim \text{Bin}(M, q)$ respectively. We can then compute the posterior probability density functions of $p$ and $q$ conditioned on the observations $\mathcal{N} = n$ and $\mathcal{M} = m$ using exact Bayesian inference.

$$f_{p|n}(x) := \mathbb{P}[p = x | \mathcal{N} = n] \tag{37}$$

$$= \mathbb{P}[\mathcal{N} = n | p = x] \underbrace{\mathbb{P}[p = x]}_{=1} \left( \int_0^1 \mathbb{P}[\mathcal{N} = n | p = x] dx \right)^{-1} \tag{38}$$

$$= \binom{N}{n} x^n (1-x)^{N-n} \left( \int_0^1 \binom{N}{n} x^n (1-x)^{N-n} dx \right)^{-1} \tag{39}$$

$$= x^n (1-x)^{N-n} \left( \underbrace{B_x(n+1, -n+N+1)}_{\text{incomplete beta function}} \Big|_{x=0}^{x=1} \right)^{-1} \tag{40}$$

$$= x^n (1-x)^{N-n} \left( \underbrace{\frac{n!(N-n)!}{(N+1)N!}}_{x=1} - \underbrace{0}_{x=0} \right)^{-1} \tag{41}$$

$$= \binom{N}{n} x^n (1-x)^{N-n} (N+1) \tag{42}$$

The integration in Equation 39 is solved using the definition of the incomplete beta function.

Without loss of generality we may also find $f_{q|m}$

$$f_{q|m}(x) = (M+1) \binom{M}{m} x^m (1-x)^{M-m} \tag{43}$$

Given these closed form posterior distributions for $p$ and $q$ we may compute the probability that the true value of $q$ is greater then $p$

$$\mathbb{P}[q > p | \mathcal{N} = n, \mathcal{M} = m] = \int_{y > x} f_{q|m}(y) f_{p|n}(x) dy dx \tag{44}$$

$$= \int_0^1 \int_x^1 f_{q|m}(y) f_{p|n}(x) dy dx \tag{45}$$

$$= \binom{M}{m} \binom{N}{n} (1+M)(1+N) \int_0^1 (1-x)^{-n+N} x^n \int_x^1 (1-y)^{-m+M} y^m dy dx \tag{46}$$

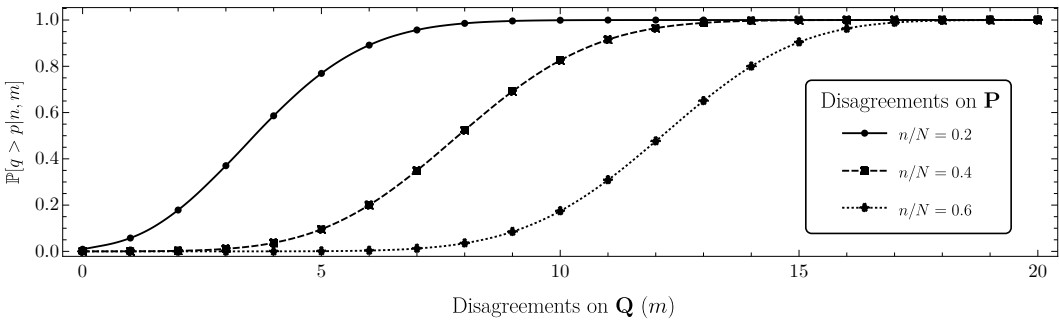

Figure 7: Belief that the probability $q$ that two classifiers disagree on samples from $\mathcal{Q}$ is greater then the probability $p$ that they disagree on $\mathcal{P}$ given an observation of $m$ disagreements out of $M$ samples from $\mathcal{Q}$ and $n$ of $N$ disagreements on $\mathcal{P}$. We plot this probability for $M = 20$ and $N = 10000$. We observe that even for a small test set size, $M = 20$, we can strongly believe that there is a true difference when $m/M$ is only slightly larger then $n/N$.

This integral, while daunting, can easily be solved in closed form using the free online Wolfram Mathematica cloud (result link). The solution is exactly Equation 34.

$$\mathbb{P}[q > p | \mathcal{N} = n, \mathcal{M} = m] = 1 - \frac{(M+1)!(N+1)!(m+n+1)!}{(m+1)!n!(M-m)!(m+N+2)!} \times$$
$$_3F_2(m+1, m-M, m+n+2; m+2, m+N+3; 1) \tag{47}$$

To gain intuition, a graphical representation of Equation 47 is provided in Figure 7. From a practical standpoint, if a practitioner trains two classifiers $f$ and $g$ that appear to disagree more often on a new dataset than a baseline rate computed on an in-domain test set, they can decide to act on that observation. (e.g., collected more training data) at a particular belief threshold (e.g., probability greater than 80%). Furthermore, there is a natural link between Equation 47 and covariate shift as exact knowledge of $\mathbf{q} > \mathbf{p}$ by definition implies not only covariate shift $\mathcal{P} \neq \mathcal{Q}$, but a type that is harmful by our original definition in section 3. Therefore knowing the probability that $q > p$ is a useful measure of how likely, without any additional assumptions, that we are experiencing a harmful covariate shift.

□

## D  CONSTRAINED DISAGREEMENT LEARNING

We elaborate on the technical details of the objective functions required for training constrained disagreement classifiers.

### D.1  DISAGREEMENT CROSS ENTROPY

**Intuition.**  In our methodology, we propose the *disagreement cross entropy* (DCE) as a simple loss function that encourages a classifier to disagree with a target label while otherwise outputting a high entropy prediction. DCE is equivalent to simply flipping the target label and computing the regular cross-entropy in the binary classification case.

Consider a model that outputs a distribution $\{p_1, p_2, 1 - p_1 - p_2\}$ over 3 classes, suppose we would like to train it to **not** output high probability for class 3 i.e minimize $p_3 = 1 - p_1 - p_2$. An intuitive approach would be to *maximize* the standard cross entropy loss $-\log(1 - p_1 - p_2)$ that one would equivalently *minimize* if trying to output high probability for class 3. We show in Figure 8 that this objective is unstable whereas the DCE $1/2(\log(p_1) + \log(p_2))$ is convex, and takes on values in a similar range to the regular cross entropy.

**Loss Function.**  Let $\hat{y}$ denote a discrete distribution over $N$ classes and $t$ a target class $t \in \{1, ..., N\}$. We define the disagreement cross entropy $\hat{\ell}$ as the cross entropy of $\hat{y}$ with the uniform distribution

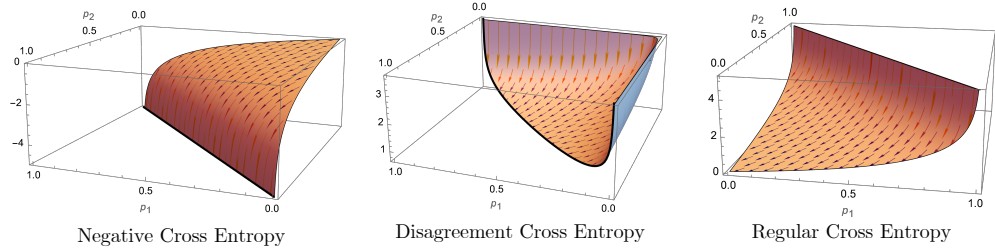

Figure 8: Consider a classifier outputs a distribution $\{p_1, p_2, 1 - p_1 - p_2\}$ over 3 classes. We compare the optimization landscape induced by either (left) minimizing the negative cross entropy for the target $p_3 = 1 - p_1 - p_2$ (e.g trying not to predict class 3) (center) minimizing our *disagreement cross entropy* (DCE) for the same target and (right) minimizing the regular cross entropy (e.g trying to predict class 3). We observe that naively minimizing the negative cross entropy results in an unbounded local minima, while the DCE is significantly more stable and scales similarly to the regular cross entropy. Gradients are overlaid to help better visualize the 3D geometry.

over all classes except $t$.

$$\hat{\ell}(\hat{y}, t) := \frac{1}{1 - N} \sum_{i=1}^{N} \mathbb{1}_{i \neq t} \log(\hat{y}_i) \tag{48}$$

This expression has the effect of minimizing the predicted probability of the target class $\hat{y}_t$ while maximizing the overall entropy of the prediction. We note that if $N = 2$ as in binary classification, then Equation 48 falls back to simple label flipping. Furthermore if we let $\hat{y} = \text{Softmax}(l)$ for some real valued $N$ dimensional logit vector $l$

$$\hat{\ell}(l, t) = \frac{1}{N-1} \left( \sum_{i=1}^{N} \mathbb{1}_{i \neq t} l_i \right) + \log \left( \sum_{i=1}^{N} \exp(l_i) \right) \tag{49}$$

When learning constrained disagreement classifiers using DCE, we minimize the loss function in Equation 3, which applies the regular cross-entropy to the in-distribution samples $\mathbf{P}$ and the DCE to the possibly shifted target samples $\mathbf{Q}$. Losses are combined with a weighted sum whose weight should be set to $1/(|\mathbf{Q}| + 1)$ as discussed in section 3 (*Choosing $\lambda$*).

**Validity of the DCE Measure.** We introduce a dentition of a *a valid disagreement loss function* and show that the DCE loss satisfies it.

**Definition 1** (A Valid Disagreement Loss Function). Let $P$ be the set of all probability vectors of length $N$ whose maximum index is a fixed target label $y \in \{1, \ldots, N\}$ . Similarly, let $Q$ be the set of all probability vectors whose maximum index is **not** uniquely equal to $y$. For instance, in the three-dimensional case with $y = 3$

$$P = \{(p_1, p_2, 1 - p_1 - p_2) \text{ s.t. } | \max(p_1, p_2) < 1 - p_1 - p_2\}$$
$$Q = \{(p_1, p_2, 1 - p_1 - p_2) \text{ s.t. } | \max(p_1, p_2) \geq 1 - p_1 - p_2\}$$
$$\text{and } 0 \leq p_1, p_2 \leq 1 \wedge p_1 + p_2 \leq 1$$

A score $S(\mathbf{p}; y)$ is proper for disagreement if

$$\forall \, \mathbf{p} \in P \, \min_{\mathbf{q} \in Q} S(\mathbf{q}; y) \leq S(\mathbf{p}; y) \tag{50}$$

which is to say that for every probability vector in $P$ there exists a probability vector in $Q$ that achieves a score at least as low.

**Theorem 3.** The Disagreement Cross Entropy Loss is a valid disagreement loss function.

*Proof.* We show that the minimum DCE for a probability vector $\mathbf{q} \in Q$ is $\log(N - 1)$ while the minimum DCE for $\mathbf{p} \in P$ is $\log(N)$. First note that by definition of the DCE in Equation 48 the unique probability vector $\mathbf{v}$ that globally minimizes it with respect to a target class $y$ is the vector

with elements $1/(N-1)$ for all indices $\mathbf{v}_i$ where $i \neq y$ and $\mathbf{v}_y = 0$. $\mathbf{v}$ is a member of $Q$ because it does not have a unique maximum at position $y$. Computing the DCE of $\mathbf{v}$ with respect to the target $y$ gives:

$$\text{DCE}(\mathbf{v}, y) = \frac{1}{1-N} \sum_{i=1}^{N} \log(\mathbf{v}_i) \mathbb{1}_{i \neq y} = \frac{1}{1-N} \log\left(\frac{1}{N-1}\right) \times (N-1) \tag{51}$$

$$= \log(N-1) \tag{52}$$

Next note that since $\mathbf{p}_y$ does not contribute to minimizing the DCE which is in the form $\sum_{p \in \mathbf{p} \backslash \mathbf{p}_y} \log(p)$ the minimal solution must minimize the term $\mathbf{p}_y$. However to enforce $\mathbf{p} \in P$ we must have that $\forall\, i \in \{1, \ldots, N\} \setminus y,\ \mathbf{p}_y > \mathbf{p}_i$. Hence minimizing $\mathbf{p}_y$ while making sure $\mathbf{p} \in P$ enforces $\mathbf{p}_y > 1/N$ for some $\epsilon > 0$ and all other $p_i < 1/N$ since if $\mathbf{p}_y \leq 1/N$ we could also chose some $p_i \geq 1/N$ and produce a vector $\mathbf{p} \in Q$. Without loss of generality choosing $\mathbf{p}_y = 1/N + \varepsilon$ for some small $\varepsilon > 0$ we must chose all other $\mathbf{p}_i = 1/N - \epsilon_i$ s.t. all $\epsilon_i > 0$ and $\sum_{i \in \{1, \ldots, N\} \setminus y} \epsilon_i = \varepsilon$. This gives a DCE of:

$$\text{DCE}(\mathbf{p}, y) = \frac{1}{1-N} \sum_{i=1}^{N} \log(\mathbf{p}_i) \mathbb{1}_{i \neq y} = \frac{1}{1-N} \sum_{i \in \{1, \ldots, N\} \setminus y} \log\left(\frac{1}{N} - \epsilon_i\right) \tag{53}$$

$$\text{where } \lim_{\varepsilon \to 0} \frac{1}{1-N} \sum_{i \in \{1, \ldots, N\} \setminus y} \log\left(\frac{1}{N} - \epsilon_i\right) = \log(N) \tag{54}$$

For all vectors $\mathbf{p} \in P$ we cannot achieve a DCE of less than $\log(N)$ but the minimum DCE in $Q$ is $\log(N-1)$ which is less than $\log(N)$, hence proving that the DCE is a suitable scoring rule for disagreement. $\qquad\square$

**Implementation.** We provide a simple PyTorch style implementation for batched computation of the CDC objective in Equation 3 given batch of logits and targets. We assume `logits` is a floating point vector of (batch_size $\times\ N$) logit values, `targets` is a integer vector (batch_size) of target classes, `mask` is a 0-1 mask of (batch_size) that is 1 at index $i$ if `logits[i]` is $\mathbf{P}$ and 0 otherwise.

```python
import torch, import torch.nn.functional as F
def cdc_loss(logits, targets, mask, weight=1/(size of test set + 1)):
    # select data in p
    logits_p, targets_p = logits[mask], targets[mask]
    # select data in Q
    logits_q, targets_q = logits[1-mask], targets[1-mask]
    # cross entropy on P with targets
    loss_p = F.cross_entropy(logits_p, targets_p, reduction=None)
    # DCE on Q
    zero_hot = 1 - F.one_hot(targets_q, num_classes=N)
    loss_q = (logits_q * zero_hot).sum(dim=1) / (1 - N)
        + torch.logsumexp(logits_q)
    # Weighted mean
    return torch.cat([loss_p, weight * loss_q]).mean()
```

## D.2 EXTENSION TO DISCRETE MODELS

When training models with arbitrary discrete or non-differentiable parameters concerning their objective (e.g., random forest), we must find a more general solution for creating CDCs. Such a solution should (1) reduce to the DCE when the model is, in fact, continuous and trained using the standard cross-entropy, and (2) reduces to label flipping when $N = 2$ (binary classification). Our simple solution is to replicate every sample in $\mathbf{Q}$ exactly $N-1$ times and create a unique label for each from the set $\mathcal{S} := \{1, \ldots, N\} \setminus \{t\}$ where $t$ is the disagreement target. We also each a sample a weight of $1/(N-1)$. In the case of $N = 2$, this corresponds to no replication and simply assigning the opposite label. In the case where the model learns by cross-entropy, it equals Equation 3.

*Proof.* We prove this statement starting with the definition of the cross entropy

$$\text{CE}(\hat{y}, y) = \sum_{c=1}^{N} \mathbb{1}_{c=y} \log(\hat{y}_c) \tag{55}$$

Now we consider the sum of the cross entropy for each label in $\mathcal{S}$:

$$\sum_{y \in \mathcal{S}} \text{CE}(\hat{y}, y) = \sum_{y \in \mathcal{S}} \sum_{c=1}^{N} \mathbb{1}_{c=y} \log(\hat{y}_c) \tag{56}$$

$$= \sum_{c=1}^{N} \mathbb{1}_{c \neq t} \log(\hat{y}_c) \tag{57}$$

$$= (N-1)\text{DCE}(\hat{y}, y) \tag{58}$$

Hence when giving each sample a weight of $(N-1)^{-1}$ we recover the exact form of DCE. $\qquad\square$

### D.3 DISAGREEMENT WITH OVERPARAMETERIZED MODELS

Many existing tests for covariate shift that rely on overparameterized models do not perform well in the small sample regime due to catastrophic overfitting. In this section, we explain how this phenomenon, if ignored, can also be catastrophic for the Detectron but is easily fixable with a simple early stopping technique. We recall the main hypothesis on which the Detectron is built:

> *Given a base classifier $f \in F$ that is well fit to a training dataset $\mathbf{P}$, it is easier to learn another classifier $g \in F$ that disagrees with $f$ on unlabeled data $\mathbf{Q}$, but agrees with $f$ on $\mathbf{P}$ when the distribution of $\mathbf{Q}$ is far from that of $\mathbf{P}$.*

Restated with the ideas introduced in this work, the above translates to the CDC objective being more easily satisfied when there is a harmful shift. This hypothesis is exemplified in Figure 9 to the right, where we see that the CDC disagreement rate grows significantly more rapidly when $\mathbf{Q}$ is chosen to be out-of-distribution (blue curve) versus in distribution (black curve). However, as the models are overparameterized, the disagreement rate eventually reaches 1 independently of whether we use the true OOD data $\mathbf{Q}$ or the ID data $\mathbf{P}^\star$. Since our test is computed based on the disagreement rate $\phi_{\mathbf{Q}}$ falling above $1 - \alpha$ quantile of the calibration distribution of $\phi_{\mathbf{P}}$ the test will be highly uninformative if $\phi_{\mathbf{Q}} = \phi_{\mathbf{Q}} = 1$.

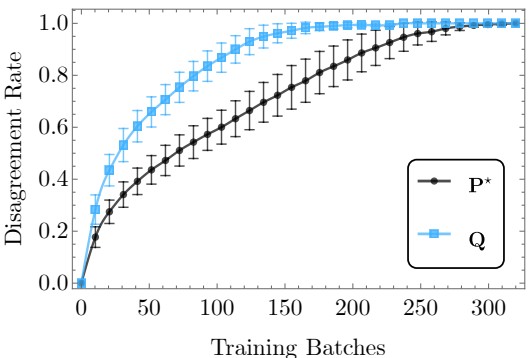

Figure 9: **CDC Training Dynamics for Overparameterized Models:** Using the same experimental setup as the runtime study in Figure 4 we record the associated CDC disagreement rate on both unseen in distribution data $\mathbf{P}^\star$ (CIFAR 10) and a known covariate shift $\mathbf{Q}$ (CIFAR 10.1). When we trained such an overparameterized model for too long the disagreement rate approaches 1 on both in and out-of-distribution datasets leading to a maximally uninformative test; it is, therefore, crucial to perform early stopping at or before the out-of-distribution disagreement rate reaches 1.

Note that it is critical to use the exact same training algorithm on both the given test set $\mathbf{Q}$ and the in-distribution calibration set $\mathbf{P}^\star$ to ensure the statistical soundness of the Detectron, so one can tune the CDC learning algorithm on either $\mathbf{Q}$ or $\mathbf{P}^\star$ to achieve a desired result then apply the exact same algorithm to the other set and test for statistical differences. In our experiments with overparameterized models, we picked a relatively low number of training iterations to use (see subsection E.2 for details) practically eliminating the issue of overfitting. However, in general, one should fix a training budget for CDCs so that they are far from reaching an ID disagreement rate of $\phi_{\mathbf{P}} = 1$. Knowing the exact level of $\phi_{\mathbf{P}}$ to stop at will depend on the dataset and learning algorithm used but as a general guideline we suggest $\approx 0.5$ to allow for a large gap between ID and OOD disagreement rates while giving enough of a compute budget to achieve non-trivial results.

# E   EXPERIMENTAL DETAILS

## E.1   BASE CLASSIFIERS

For each dataset used in our experiments, we begin by training a *base classifier* on the source domain portion of the dataset to use in subsequent experiments and baselines. For a brief description of the datasets used and base classifiers, see Table 4 and for a more detailed description of each dataset as well as what we have considered precisely as the source and shifted domains, see subsection F.2.

**CIFAR 10.**   We use a standard Resnet18 model pre-trained on ImageNet [Deng et al., 2009] made available in the torchvision library [Marcel and Rodriguez, 2010] although we reinitialize the last network layer to have an output size of 10. We use stochastic gradient descent (SGD) with a base learning rate of 0.1, $L_2$ regularization of $5 \times 10^{-4}$, momentum of 0.9, a batch size of 128 and a cosine annealing learning rate schedule with a maximum 200 iterations stepped once per epoch for a total of 200 epochs. We use the standard CIFAR-10 training split normalized by its mean ($\mu = [0.4914, 0.4822, 0.4465]$) and standard deviation ($\sigma = [0.2023, 0.1994, 0.2010]$). Every epoch, we randomly crop each image to a size of $32 \times 32$ after applying a 0 padding of four pixels to each spatial dimension, and we apply a horizontal flip with probability 0.5. This model achieves a test performance of 87%. While this score is far from state-of-the-art on CIFAR-10, our goal is not to construct a perfect model. We wish to create a *reasonably good* model as an example of a model that could realistically be deployed in real-world settings. When training deep ensembles, we only vary the random seed in the range $[0, \ldots, 4]$.

**Camelyon 17.**   We follow a similar approach to CIFAR 10. However, we use two output features (for binary classification of cancerous or benign pathology), a batch size of 512, the ADAM optimizer [Kingma and Ba, 2015] with a base learning rate of 0.001, $L_2$ regularization of $10^{-5}$ and a total of 5 training epochs for which we select the model with the best validation accuracy. This model achieves a test accuracy of 0.93. When training deep ensembles, we only vary the random seed in the range $[0, \ldots, 4]$.

**UCI Heart Disease.**   We train both neural networks and gradient boosted trees using the XGboost library [Chen and Guestrin, 2016]. For the neural network model, we use a simple MLP with an input dimension of 9, 3 hidden layers of size 16 with ReLU activation followed by a 30% dropout layer and a linear layer to 2 outputs (heart disease present or not). We use 358 samples for training and 120 for validation. We train for a maximum of 1000 epochs and select the model with the highest AUC on the validation set, performing early stopping if the validation AUC has not increased in over 100 epochs. This model achieves a test AUC computed on 119 samples of 0.85. As with CIFAR 10 and Camelyon 17, we only vary the random seed in the range $[0, \ldots, 9]$ when training deep ensembles. Note that we chose a larger ensemble size here as models are fairly cheap to train. Another important trick when using small $\mathbf{Q}$ sizes is to sample all of $\mathbf{Q}$ in each batch filling the best with a random set of samples from $\mathbf{P}$. Of procedure artificially inflates the size of $\mathbf{Q}$ so the hyperparameter $\lambda$ must account for this by picking up an extra multiplicative factor equal to (batches per epoch)$^{-1}$.

When training gradient boosted trees using XGboost we employ standard library parameters ($\eta = 0.1$, eval_metric=auc, max_depth= 6, subsample= 0.8, colsample_bytree=0.8, min_child_weight= 1, objective=binary:logistic, num_round= 10). This model while taking less then $5s$ to train achieves a test AUC of 0.88.

## E.2   CONSTRAINED DISAGREEMENT CLASSIFIERS

We expand on the experimental details for learning constrained disagreement classifiers (**??**). When training a CDC $g_{(f, \mathbf{P}, \mathbf{Q})}$ we start by creating a new dataset that combines all elements of the labeled set $\mathbf{P}$ and the unlabeled set $\mathbf{Q}$ with pseudo labels inferred by the base classifier $f$. We store a single bit for each sample in the combined dataset to indicate if a sample was originally drawn from $\mathbf{P}$ or $\mathbf{Q}$. When training CDCs with neural networks, we use the DCE loss ($Equation$ 3) under similar semantics as the pseudo-code implementation provided above. When training discrete models, we resort to our generalized approach in subsection D.2. To reduce training time we initialize $g$ using the exact same architecture/weights as $f$ and apply the exact same optimization algorithm/learning rate used to train $f$ (see subsection E.1). For CIFAR 10, we train each CDC for a maximum of 10 epochs performing early stopping if the model drops in in-distribution validation performance by over 5%.

We enforce the early stopping criteria to help prevent CDCs from overfitting to the disagreement loss when the target dataset has not come from a harmfully shifted domain. The intuition is the following: under the null, if a target dataset $\mathbf{Q}$ comes from the same distribution as a training dataset $\mathbf{P}$, then learning to disagree with $f$ on $\mathbf{Q}$ while constrained to agree on all of $\mathbf{P}$ can only be solved by overfitting to predict with high entropy on the specific examples in $\mathbf{Q}$, versus learning a distinct pattern that distinguishes the distributions. Forcing a model to predict with high entropy on a subset of in-distribution datapoints can only hurt its associated in-distribution generalization, a phenomenon which we can directly assess by measuring validation performance.

The details for training CDCs on Camelyon 17 are the same as those described for CIFAR 10, however due to the large training set size ($302436$ samples) we simply select a random subset of size $50,000$ as $\mathbf{P}$ at each epoch – a number we experimentally deemed as sufficient to achieve low in-distribution generalization error. When training CDCs on the UCI Heart Disease dataset, we use XGBoost [Chen and Guestrin, 2016] with the same hyperparameters described in subsection E.1.

For the runtime experiment presented in Figure 4 we train each CDC for only one batch, where each batch contains a set of 100 samples $\mathbf{Q}$ and is filled up to a batch size of 512 with random samples from $\mathbf{P}_{\text{train}}$. After every batch we eliminate all samples where the CDC disagrees with the base predictions. We continue this for a maximum of 150 batches, but perform early stopping if 10 batches pass without at least one sample getting disagreed on.

### E.3 DESCRIPTION OF BASELINE METHODS

We compare the Detectron against several methods for OOD detection, uncertainty estimation and covariate shift detection found in recent literature.

1. *Deep Ensembles* shown by Ovadia et al. [2019] to provide the most accurate estimates of predictive confidence under covariate shift. To compare directly with Detectron we test both the disagreements rates and the entropy distributions of the ensemble. See Appendix B for more information on how these tests are run.

2. *Black Box Shift Detection (BBSD)* [Lipton et al., 2018] is overall best method across numerous synthetic benchmarks for covariate shift detection evaluated by Rabanser et al.. We follow the same evaluation and perform a univariate KS test on each dimension of the softmax output of the base classifier between $\mathbf{Q}$ and a held out set from the training distribution. Bonferroni correction is used to compute a single $p$-value as the minimum value divided by the number of tests. We guarantee significance using the same permutation approach described in Appendix B.

3. *Relative Mahalanobis Distance (RMD)* [Ren et al., 2021] (a method designed specifically for identifying near OOD samples) using the penultimate layer of a pretrained model. We test for covariate shift by performing a KS test directly on the distribution of RMD confidence scores derived on $\mathbf{Q}$ and $\mathbf{P}^{\star}$.

4. *Classifier two sample test (CTST)* [Lopez-Paz and Oquab, 2017]. Using the same architecture as and initialization as the base classifier we reconfigure the output layer and we train a domain classifier on half the test data with source data labeled as 0 and test data as 1. We then test this models accuracy on the other half of the test data and compare its performance to random chance using a binomial test (see Appendix B for more details). While this method is technically sound it is not suitable for the low data regime where learning a domain classifier on half the test data is unlikely to generalize beyond random performance on the other half.

5. *Deep Kernel MMD* [Liu et al., 2020]. We use the authors original source code available at `https://github.com/fengliu90/DK-for-TST` to perform the deep kernel MMD test.

6. *H-Divergence* [Zhao et al., 2022]. Most similar to our approach, this work proposes a two sample test based on the output of a learning model after training on either source or target data. Specifically, the authors fit a model to both the source dataset $\mathcal{P}$, the target dataset $\mathcal{Q}$ and a uniform mixture $(\mathcal{P} + \mathcal{Q})/2$. Under the null hypothesis $\mathcal{P} = \mathcal{Q}$ the loss in each case is equal in expectation. However when $\mathcal{P} \neq \mathcal{Q}$, the generalized entropy of the mixture distribution may be be larger. In practice the authors fit three VAE [Kingma and Welling,

2014] models and compute the test statistic $\ell((\mathcal{P} + \mathcal{Q})/2) - \min(\ell(\mathcal{Q}), \ell(\mathcal{P}))$, where $\ell$ is the VAE loss computed as a sum of the binary cross entropy reconstruction loss and the KL divergence regularizer. The perform 100 runs where the null hypothesis (e.g. sample $\mathbf{Q}$ from $\mathcal{P}$) and one where it does not. Significance is determined in the standard way be observing if the true test statistic exceeds the $95^{\text{th}}$ percentile of the test statistic distribution under the null hypothesis. Unfortunately this method, while state of the art on several benchmarks including the MNIST vs Fake MNIST two sample test, demonstrated low utility on more complex tasks with smaller sample sizes. After a discussion with the authors, we attempted to improve the results by first pretraining the VAE to produce valid samples and reconstructions under the source distribution and computing the H-Divergence statistic after finetuning. Despite this effort, we still was low statistical significance with small sample sizes likely due to the noisy nature of training VAE's in the low data regime. We use the authors original source code available here `https://github.com/a7b23/H-Divergence`.

### E.4 Comparison to Generalization Error Prediction

Related to distribution shift detection is the problem of estimating out of distribution generalization error on unlabeled data. Any method that estimates generalization error can be thought of as a regression from a dataset on a real valued test statistic $\psi$, which represents the estimated generalization error. As such we can calibrate the distribution of $\psi$ based on unseen iid data in $\mathbf{P}^{\star}$ and test for distribution shift by determining if the predicted generalization error on $\mathbf{Q}$ is within the $5\%$ extreme of the calibration distribution.

Many recent methods [Platanios et al., 2016; Yu et al., 2022; Garg et al., 2022] have been proposed to address this problem. While Garg et al. propose the simplest approach their method is conceptually identical to BBSD when applied to the task of shift detection. Hence we compare only with the Projection Norm approach of Yu et al. as it presents the current state of the art and provides a well documents experimental repository. A comparison on the CIFAR10/10.1 benchmark is found in Table 2 and a further analysis of the scaling is given in Table 3. Ultimately, the Projection Norm presents another useful approach for identifying covariate shift, however the computational complexity is at least equal or in most cases greater than the Detectron and the performance is in all cases lower.

Table 2: Comparison of Detectron (Entropy) and the Projection Norm [Yu et al., 2022]. We report the TPR@5 for both methods using sample sizes of 10, 20 and 50.

|  | $|\mathbf{Q}| = 10$ | $|\mathbf{Q}| = 20$ | $|\mathbf{Q}| = 50$ |
| --- | --- | --- | --- |
| ProjNorm [Yu et al., 2022] | $.11 \pm .03$ | $.17 \pm .04$ | $.39 \pm .05$ |
| Detectron (Entropy) | $\mathbf{.35 \pm .05}$ | $\mathbf{.56 \pm .05}$ | $\mathbf{.92 \pm .03}$ |

Table 3: Projection Norm results [Yu et al., 2022] on larger samples sizes of CIFAR10/10.1. We observe that $\approx 500$ samples are required to reach near perfect test power.

|  | $|\mathbf{Q}| = 100$ | $|\mathbf{Q}| = 200$ | $|\mathbf{Q}| = 500$ | $|\mathbf{Q}| = 1000$ |
| --- | --- | --- | --- | --- |
| ProjNorm [Yu et al., 2022] | $.48 \pm .05$ | $.76 \pm .04$ | $.99 \pm .01$ | $1.0 \pm 0$ |

## F Datasets

### F.1 Sources and Licensees

- CIFAR-10 (MIT License Copyright (c) 2013 Valay Shah)
- Camelyon-17 (CC0 1.0 Universal Public Domain Dedication)
- UCI Heart Disease (Creative Commons Attribution 4.0 International)

## F.2 PREPOSSESSING, SHIFT DESCRIPTIONS AND MODEL PERFORMANCE

We provide full details on the three datasets used in our experiments, including any preprocessing steps and what splits we considered as source domain $\mathcal{P}$ and target domain $\mathcal{Q}$.

**CIFAR 10/10.1.** We use the well known CIFAR 10 dataset [Krizhevsky et al., 2014] as the source domain for training base classifiers (subsection E.1) and the new CIFAR 10 test set CIFAR 10.1 containing 2000 class balanced images [Recht et al., 2019] as a source of harmful distribution shift. Although the images in CIFAR 10.1 appear to be visually very similar to CIFAR 10 most classifiers trained on CIFAR 10 drop significantly in performance (3% to 15% [Recht et al., 2019]) when tested on CIFAR 10.1.

Figure 10: Cifar 10 vs Cifar 10.1. (Image borrowed from the technical report "Do CIFAR-10 Classifiers Generalize to CIFAR-10?" [Recht et al., 2018])

**Camelyon 17.** As described by the original authors [Veeling et al., 2018] the Camelyon benchmark is a new and challenging image classification dataset consisting of 327,680 color images ($96 \times 96$) extracted from histopathologic scans of lymph node sections. Each image is annotated with a binary label indicating the presence of metastatic tissue in the center $32 \times 32$ pixel region. In our experiments, we use the WILDS [Koh et al., 2021] framework to facilitate download, preprocessing, as well as source/target splits for Camelyon. As shown in Figure 11, the source domain is chosen as data from hospitals $1, 2, 3$. In contrast, the test domain is collected from hospital $5$, which visually shows significantly higher contrast due to different data acquisition equipment/methods.

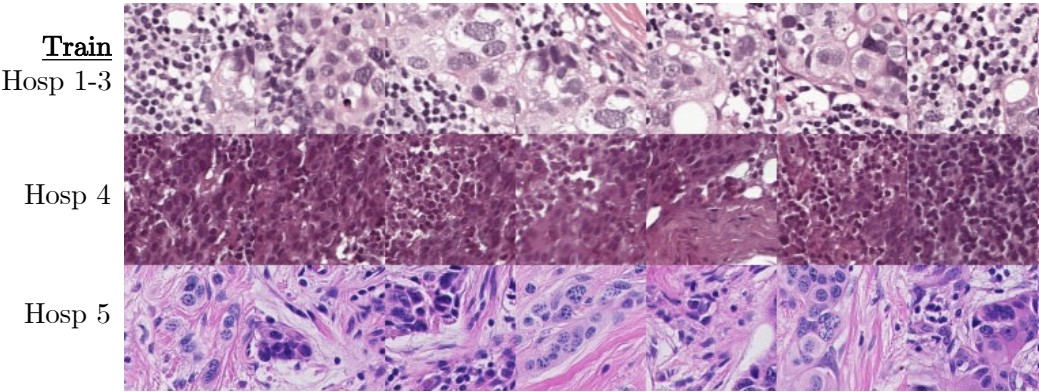

Figure 11: Samples images from the Camelyon 17 dataset [Veeling et al., 2018]. Using the standard set by the WILDS framework [Koh et al., 2021] we use hospitals 1-3 as the source domain for training and validating models, and hospital 5 as the target domain for assessing distribution shift.

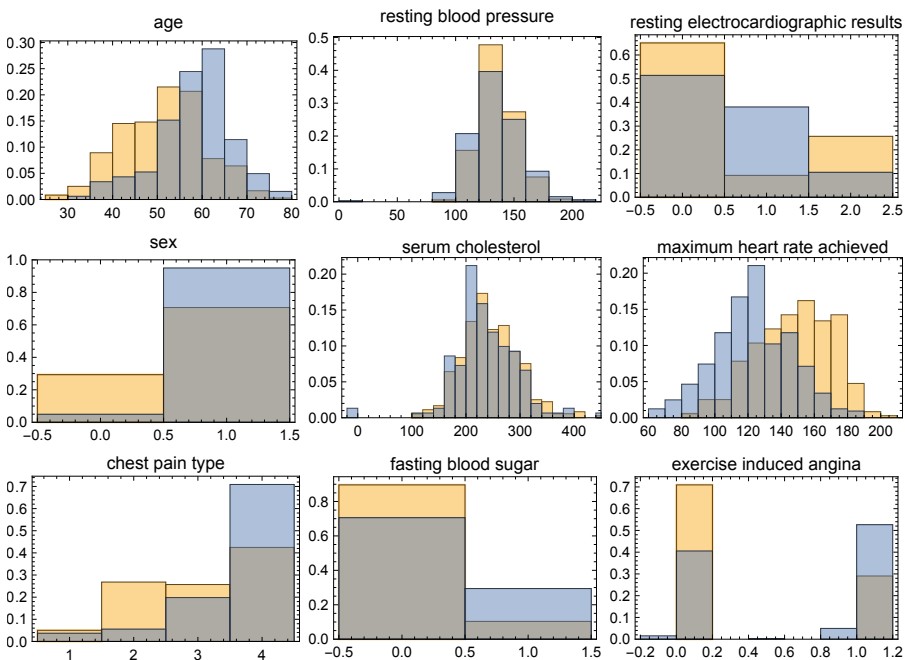

Figure 12: Marginal distributions for each of the nine variables chosen from the UCI Heart Disease dataset for our experiments. The source domain (yellow) is chosen from the Cleveland and Hungary patient databases, while the shifted target domain (blue) is selected from the Swiss and Long Beach databases. Although the distributions are visually similar, a simple neural network classifier that achieves an AUC of 0.85 on the source domain drops to 0.42 on the target domain.

**UCI Heart Disease.** The UCI Heart Disease (UCI-HD) dataset [Janosi et al., 1988] consists of 76 attributes collected from four unique patient databases in Cleveland, Hungary, Switzerland, and the VA Long Beach. We select nine features out of the commonly used 14 to minimize the portion of missing values. These features are {age, sex, chest pain type, resting blood pressure, serum cholesterol, fasting blood sugar, resting electrocardiographic results, maximum heart rate achieved, exercise-induced angina}. The prediction task is to determine the diagnosis of heart disease (also known as angiographic disease status), which is given in a range from 0-4, where 0 indicates healthy and 1-4 indicates a severity level based on the narrowing of major blood vessels. Following prior work [Chaki et al., 2015] we only consider the simplified binary classification task for differentiating patients with a normal angiographic status (label of 0) from those with abnormal status (label $> 0$). We select the source domain as the Cleveland and Hungary databases and the target domain as the Switzerland and VA Long Beach databases. A graphical overview of the marginal feature distributions for the source and target domain is shown in Figure 12.

To allow for out-of-the-box training of deep neural networks on the UCI HD dataset, we use the missing value synthesis functional in Wolfram Mathematica [Inc.]. The algorithm uses density estimation and mode finding on conditioned distributions to synthesize missing values. See the language guide page titled Synthesize Missing Values in Numeric Data for a more detailed description. To aid in future research, we provide a copy of our processed dataset in `<our github repo>/data/uci_heart.pt`.

A summary of the three datasets used as well as the description of shifts and effects on model performance is provided in Table 4.

## G  EXPANDING ON GENERALIZATION REGIONS

We expand on the concept of the generalization region $\mathcal{R}$ proposed in section 3. We highlight that not all covariate shifts will be harmful to the performance of a model, as in many cases, the model will generalize to a region more extensive than the support of the training distribution. We informally

Table 4: **Datasets:** We investigate three different forms of covariate shift. To verify that these shifts are indeed harmful to the models, we report performance in both the shifted and unshifted domains. Examples and further descriptions of unshifted/shifted splits of each dataset are given in Appendix F.

| Domain/Task | Dataset | Shift | Metric | (Unshifted) | (Shifted) |
|---|---|---|---|---|---|
| Natural Images *Object classification* | CIFAR-10/10.1 [Recht et al., 2019] | Data Collection Process | Accuracy | 0.87 (Resent18) | 0.77 (Resent18) |
| Histopathological Images *Metastases Detection* | Camelyon-17 [Veeling et al., 2018] | Different Hospitals | Accuracy | 0.93 (Resent18) | 0.81 (Resent18) |
| Tabular Medical Data *Angiographic Status* | UCI Heart Disease [Janosi et al., 1988] | Different Countries | AUROC | 0.88 (xgboost) 0.85 (MLP) | 0.70 (xgboost) 0.42 (MLP) |

Figure 13: We train a CNN with the LeNet-5 architecture [LeCun et al., 1989] on a dataset consisting of MNIST images rotated from $0$ to $80$ degrees counterclockwise in steps of $20$. We test the model using rotations of $-25$ to $125$ degrees in steps of $5$. We observe that the model achieves nearly identical test accuracy on all test angles between $-5$ and $85$, indicating that it has generalized to angles outside of its training set.

refer to this region as $\mathcal{R}$ and note that characterizing it precisely is intractable as it depends on the complex interaction between a model, dataset, and learning algorithm. When detecting distribution shifts, we are primarily concerned with those that will harm the performance of a predictive model in deployment; those are shifts that are not simply any changes in the training distribution but ones that assign a high probability to samples outside of the generalization region. The Detectron aims to use not just a model and dataset to detect shifts but also the learning algorithm to tie the detection of shifts directly with $\mathcal{R}$.

While this argument is informal, we present an experiment to show that a non-trivial generalization region does exist in a simple machine learning task. Figure 13 shows that a LeNet-5 model [LeCun et al., 1989] trained on rotated MNIST images can match in-distribution performance when tested on various rotations outside of its training set. We train this model for ten epochs using the ADAM optimizer at a base learning rate of $0.001$. By leveraging a model's existing learning algorithm when detecting shifts using the Detectron we can detect if new and unlabeled datasets are likely to belong to $\mathcal{R}$ by training a new model that is constrained to learn the same behavior over $\mathcal{R}$ while encouraging it to predict randomly otherwise. While this experiment is simplistic, it shows that learning algorithms can generalize to sets outside their training distribution.

## H   HARMFUL COVARIATE SHIFT AND CONNECTION TO $\mathcal{A}$ DISTANCE

Given a labeled training set **P** as well as another labeled dataset **Q**, one can identify using standard statistical estimation if a model $f$ performs more poorly on **Q** compared to **P**. However, in a practical scenario, decision models are deployed on unlabeled datasets; hence directly computing model

performance is impossible. To decide then if $\mathbf{Q}$ has been drawn from a distribution that may cause $f$ to fail, we formulate an adversarial learning style definition of *harmful covariate shift* that does not require access to labeled examples.

**Definition: $(\ell, \alpha, F)$-Harmful Covariate Shift.**
A covariate shift from distributions $\mathcal{P} \to \mathcal{Q}$ over $X$ is $(\ell, \alpha, F)$-harmful with respect to a set of decision models $F$, if there exists any subset of models of two or more models $\mathbf{f}$ in $F$ that achieve a source domain loss $\ell(f, \mathcal{P}) \leq \alpha$ for all $f \in \mathbf{f}$ while being more likely to disagree with each other on an unseen sample from $\mathcal{Q}$ compared to $\mathcal{P}$.

$$\exists\, \mathbf{f} \subseteq F, \text{ s.t. } \forall f \in \mathbf{f}\ \ell(f, \mathcal{P}) \geq \alpha \text{ and} \tag{59}$$

$$\mathbb{P}_{x \sim \mathcal{Q}}(\exists\, f_i, f_j \in \mathbf{f} \text{ s.t. } f_i(x) \neq f_j(x)) > \mathbb{P}_{x \sim \mathcal{P}}(\exists\, f_i, f_j \in \mathbf{f} \text{ s.t. } f_i(x) \neq f_j(x)) \tag{60}$$

In plainer words, we define harmful covariate shift based on the existence of multiple *good* models on $\mathcal{P}$ that tend to disagree on $\mathcal{Q}$. The Detectron algorithm is designed to learn these models (constrained disagreement classifiers) and statistically test their disagreement rates.

We can connect our definition of harmfulness to the well-studied concept of $\mathcal{A}$ distance from Kifer et al. [2004]. The $\mathcal{A}$ is a generalization of the total variation to an arbitary collection of measurable events $\mathcal{A}$.

$$d_{\mathcal{A}}(\mathcal{P}, \mathcal{Q}) = 2 \sup_{A \in \mathcal{A}} |\mathrm{Pr}_{\mathcal{D}}[A] - \mathrm{Pr}_{\mathcal{D}'}[A]| \tag{61}$$

Ben-David et al. [2006] shows that when they chose a class of events whose characteristic functions are functions in $F$, the $\mathcal{A}$ distance in connection with VC theory [Vapnik, 1995] allows for finite sample generalization bounds on the performance of arbitrary decision models from $F$ under covariate shift. Ben-David et al. [2006] go on to show that the $\mathcal{A}$ distance defined for a binary function class $F$ is equal to

$$d_F(\mathcal{P}, \mathcal{Q}) = 2\left(1 - 2\min_{f \in F} \mathrm{err}(f)\right) \tag{62}$$

where $\min_{f \in F} \mathrm{err}(f)$ is the minimum error that a domain classifier from $F$ can achieve on the task of distinguishing samples from $\mathcal{P}$ and $\mathcal{Q}$ (i.e. if $\mathcal{P} = \mathcal{Q}$ the best domain classifier will have error of $0.5$ and $d_F(\mathcal{P}, \mathcal{Q}) = 0$ and if $\mathcal{P}$ and $\mathcal{Q}$ can be perfectly discriminated by some $f \in F$ the $d_F(\mathcal{P}, \mathcal{Q})$ is maximized and equal to 2). In our characterization of harmful covariate shift, we consider not just the discriminative power of $F$ but the broader generalization region (Appendix G) induced by training $f$ to achieve a certain source domain loss on $\mathcal{P}$. For instance, if a model naturally learns rotational invariance, as shown in Figure 13, one would also want to use a shift detector that will not detect shifts that only comprise of rotations. Beyond the concept of harmfulness, we empirically show that learning to detect shifts using CDCs instead of domain classifiers improves shift detection performance.

# I ON MODEL COMPLEXITY AND THE DETECTRON

In our methodology, we aim to specifically detect covariate shifts that lead to unpredictable behavior of models over a given function class $F$. To better understand how the complexity of $F$ changes the behavior of the Detectron, we present a simple example in Figure 14.

The Detectron is designed to work in the central and right case of Figure 14 (e.g., where we learn a function class $F$ that contains the true underlying mechanism that labels our data). However, the types of shifts considered harmful will change with the complexity of $F$. In the ideal case where the model class complexity matches that of the true mechanism, we see that the model family does not allow significant variation on the decisions for points outside but close to the training distribution meaning the Detectron will not, by design, detect nearby shifts. When models are overly complex, more types of shifts should naturally be considered harmful as there is more variation than models from $F$ can possess outside of the training domain, meaning we can never guarantee that we have chosen the correct model. In the last case, where the model family is too simple to contain the true mechanism, the Detectron loses its power to identify shifts that may result in performance penalties, as would any comparable method from OOD/uncertainty estimation that relies on a sufficiently well-calibrated classifier. The inability of our approach to handle this case is an unavoidable limitation of our methodology. However, models that are simpler than their underlying mechanism will often exhibit poor held performance even on in-distribution tasks, drastically limiting the likelihood of deployment.

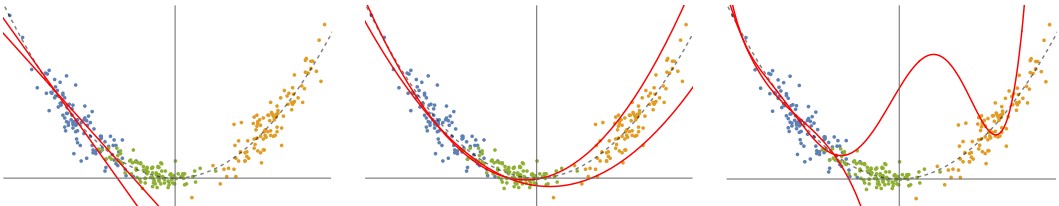

Figure 14: Investigating the relation between CDC model complexity and the ability to identify covariate shift. We consider a toy example where the ground truth labels are generated using a quadratic decision boundary, shown as a black dashed line. The blue points correspond to training samples, and the orange/green points correspond to two different covariate shifts, one closer to the training distribution and the other further. (Left) When we choose an overly simplified model family (e.g., linear classifiers), there exists no CDC that reports different explanations for the orange and green points. (Center) When we choose a quadratic function family, there exists enough variation within the space of models that explain the training set to offer different explanations on the distance shift (orange) but not on the near shift (green). (Right) When we learn from an overly expressive function family (polynomials of degree 3+), the space of models that explain the training set can offer different explanations of even near covariate shifts (green points).

In general, choosing a model family that is well specified for a given task is nontrivial and still an open question in machine learning. This concern is somewhat diminished in cases where models are over-parameterized (e.g., deep neural networks) or models are built using expert knowledge related to the true underlying mechanism, as is often the case in practical machine learning problems.

