# OpenReview forum: "A Learning Based Hypothesis Test for Harmful Covariate Shift"
_ICLR.cc/2023/Conference — ICLR 2023 poster_

### Official Review · Reviewer_C9Zo · 2022-10-24

**Confidence:** 3
**Correctness:** 3
**Technical Novelty And Significance:** 3
**Empirical Novelty And Significance:** 3
**Recommendation:** 6

**Clarity, Quality, Novelty And Reproducibility:**

- The paper is mostly clearly written, and well-supported by the Appendix.


**Strength And Weaknesses:**

Strengths:
- The authors provide very detailed theoretical and empirical analysis of their method, in both the main paper and the appendix.
- The proposed method works well, beating several other methods for detecting covariate shifts in the experiments.


Weaknesses:
- While the approach provides good empirical performance, the core idea of the method is still unclear to me, in terms of how the assumptions might fail. The main idea of the method is to detect harmful covariate shifts by training two classifiers, forcing them to agree on P while disagreeing on Q. There are several possible scenarios:
  1. If it's possible to train such a pair of classifiers f and g, it does not necessarily mean the shift from P to Q is harmful. Consider linear model on disjoint supports P and Q. It just means P and Q are different. If we just want to detect shifts between P and Q, why do we not use non-parametric tests such as the KS test, or train a logistic regression model distinguishing P and Q as commonly used in covariate shift correction?
  2. If it's not possible to train such a pair of classifiers f and g, it might just be that the models are underfitting. Consider fitting linear models to quadratic target y = x^2 between -1 and 1. P is positive region, Q is negative region. It is not possible to fit linear models that agree on P but disagree on Q, but P and Q are distinct.

  Although these are specific boundary cases on potential failure modes, the fact that the current method does not seem to cover them properly make me uncertain about the the method.


**Summary Of The Paper:**

The authors propose a method for detecting covariate shift by training two classifiers to agree on a distribution P and disagree on another distribution Q, and then build a hypothesis test on top of these two classifiers. Their proposed method is able to detect covariate shifts better than several competing approaches. However, there seems to be potential failure cases for the method that the authors do not discuss sufficiently.


**Summary Of The Review:**

This paper propose an interesting approach for detecting covariate shifts, and the authors spend a lot of effort in its theoretical and empirical analysis. However, there seems to be potential cases that the method can fail which the authors do not cover sufficiently. A better discussions on these, even listing them as limitations or assumptions, can help improve the paper.

---

> ### Author Response · Authors · 2022-11-09
> **Response from Authors: Elaborating on Corner Cases (1/2)**
>
> Thank you for bringing up these points, they are valid and were addressed in Appendix I of the original manuscript (referenced to by the main text on page 4).
>
> In general, the notion of HCS maps onto a class of shifts based on the models used to classify the training data. If the model is uniquely identified on the training set then it means that the practitioner believes they found the “right“ model  (e.g. a model based on knowledge of the causal mechanism of the functional process that relates y and x) with respect to future shifts.
>
> **Case 1: linear models + disjoint support**
>
> - If $P$ and $Q$ have partially disjoint support we may not always be able to train two linear models that correctly classify $P$ but disagree on $Q$ if the training set $P$ is sufficient to identify a unique model. For example, [here](https://difficult-kookaburra-1cd.notion.site/image/https%3A%2F%2Fs3-us-west-2.amazonaws.com%2Fsecure.notion-static.com%2Faaf76a5b-e86e-42c8-98c8-ae8604e60438%2FUntitled.png?table=block&id=179eb568-5072-44e2-9722-032f74c7d654&spaceId=d3979605-a302-4e2e-9518-41a36418134d&width=750&userId=&cache=v2): the training set (+ and - points) specifies linearly separable data (and consequently a unique linear model). There cannot be another model that agrees on the training set but gives different predictions on the unlabeled black points meaning the shift is not harmful (i.e. the practitioner believes in linear separability of the data and expects that future unlabelled points also meet this assumption).
> - If there is underspecification i.e. multiple linear models can classify the training set but have different test predictions (black points) then we say there is a harmful covariate shift. (see example [here](https://difficult-kookaburra-1cd.notion.site/image/https%3A%2F%2Fs3-us-west-2.amazonaws.com%2Fsecure.notion-static.com%2Fc493a0eb-557d-42fb-a4a6-ece31e0f3d02%2FUntitled.png?table=block&id=48fc48b1-1afb-4b17-b38d-1e72454eadb5&spaceId=d3979605-a302-4e2e-9518-41a36418134d&width=740&userId=&cache=v2)).
> - **(KS Tests and Logistic Regression)** With high dimensional data KS tests require dimensionality reduction [[Rabnasser et al](https://arxiv.org/abs/1810.11953)]. The baseline method of black box shift detection BBSD [[Lipton et al](https://arxiv.org/abs/1802.03916)] is one such method that performs a KS test on the output of a pretrained model. Similar to logistic regression we also compare to the classifier two sample test CTST [[Lopez et al](https://arxiv.org/abs/1610.06545)] which identifies shift using a deep model trained to separate source and target datasets.

---

> > ### Author Response · Authors · 2022-11-09
> > **Response from Authors: Elaborating on Corner Cases (2/2)**
> >
> > **Case 2: Quadratic Target**
> >
> > - If there is only one quadratic function that can fit a dataset, and it is known that the true hypothesis class is quadratic we are able to identify the correct model! This means that no distribution shift can be harmful. We provide a visual example [here](https://difficult-kookaburra-1cd.notion.site/image/https%3A%2F%2Fs3-us-west-2.amazonaws.com%2Fsecure.notion-static.com%2F25dc9caa-8605-4e97-9e95-401b1cdba267%2FUntitled.png?table=block&id=07609cbe-f667-4ecc-93a1-1c56a1285270&spaceId=d3979605-a302-4e2e-9518-41a36418134d&width=1060&userId=&cache=v2): a quadratic curve is used to fit a dataset of samples drawn in the region $(x,y) \in \\{-1,1\\}^2$ as there is no variability on the choice of quadratic curves that fit the dataset the model is correctly specified even under distribution shift.
> > - When there are multiple models that correctly characterize the training set we cannot identify the “correct one”  in certain regions. In this case, the shift is harmful with respect to a quadratic model class. See the example image [here](https://difficult-kookaburra-1cd.notion.site/image/https%3A%2F%2Fs3-us-west-2.amazonaws.com%2Fsecure.notion-static.com%2F5e2650dd-e56e-4540-bc6c-eb652e4cab4f%2FUntitled.png?table=block&id=e307d8c4-d96d-4b39-858f-cb053faa253d&spaceId=d3979605-a302-4e2e-9518-41a36418134d&width=1060&userId=&cache=v2).
> > - If we instead chose a linear function class, the set of models constrained to fit the dataset is not diverse enough to disagree on any test points (example  [here](https://difficult-kookaburra-1cd.notion.site/image/https%3A%2F%2Fs3-us-west-2.amazonaws.com%2Fsecure.notion-static.com%2F858b92f6-1dff-4d4b-91b6-e67b93d3704b%2FUntitled.png?table=block&id=b6182433-da3f-4d2c-94e3-2391cd2a9201&spaceId=d3979605-a302-4e2e-9518-41a36418134d&width=1060&userId=&cache=v2)). This means one of two things: (1) the shift is not harmful with respect to a linear hypothesis class and the well-calibrated shift detector will correctly not identify this shift or (2) the linear model is underspecified for the task and the potentially harmful shift will go undetected.
> > - At the end of Appendix I, we have mentioned that an unavoidable limitation of our method is that it may fail to detect shifts with underspecified models. We go on to justify that, models simpler than their underlying mechanism will often exhibit poor held performance on in-distribution tasks, drastically limiting the likelihood of real-world deployment. In general, choosing a model family that is well-specified for a given task is nontrivial and still an open question in machine learning. This concern is somewhat diminished in cases where models are over-parameterized (e.g., deep neural networks) or models are built using expert knowledge related to the true underlying mechanism, as is often the case in practical machine learning problems.
> >
> > To ensure the reader is aware of this, we **have added this point as a limitation to the conclusion in the main text [conclusion, paragraph 2]**
> >
> > Please let us know if you have any additional comments/concerns regarding this explanation and if you have any thoughts on how to additionally make these topics clear in the manuscript.

---

> > > ### Comment · Reviewer_C9Zo · 2022-12-03
> > > **Thank you for the response**
> > >
> > > I thank the authors for the detailed explanations. It is important to address some of these corner cases as they affect our choices on when to apply the method. I appreciate the inclusion of limitations of the method in the appendix. I have updated the score of my review accordingly.

---

> > > > ### Author Response · Authors · 2022-12-03
> > > > **Reply from Authors**
> > > >
> > > > Thank you for time and effort spent reviewing our work!

---

### Official Review · Reviewer_En7A · 2022-10-26

**Confidence:** 4
**Correctness:** 3
**Technical Novelty And Significance:** 3
**Empirical Novelty And Significance:** 3
**Recommendation:** 6

**Clarity, Quality, Novelty And Reproducibility:**

Clarity (4/5): This paper is mostly easy to follow, and readers with similar backgrounds should get the central concept without significant issues. Some mathematical notations might require additional marks to avoid confusion. For instance, $N$ is first used as the number of classes and then as the number of samples. $n$ denotes the dataset size at first but is then used as the number of unseen samples.

Quality (3/5): While the paper provides an interesting solution and good coverage of drift detection approaches, some technical details might require further justifications. Also, the paper doesn't cover some related work that aims to estimate the performance with an unlabelled test set.

On the technical details: (1) regarding the proposed disagreement-cross-entropy, while the authors provide some intuition, it should be aware that the cross-entropy loss and associated entropy measure both come with theoretical properties (i.e. proper scoring rules). I wonder if the authors can provide any properties of the proposed loss, as it is helpful to know whether the loss can lead to a biased disagreement (other than a pre-defined and consistent one). (2) At the beginning, I was surprised to see that the learned deep-MMD performed poorly in detecting drifts that even affect model performance. Then I realised that the experiments are performed with quite a limited sample size. In this case, I might further consider reducing the complexity of the projection net in methods like deep kernel MMD and classifier two-sample test, as it is just not practical to train a large model with such a limited sample size. (3) Following the discussion on sample size,  it is clear that the proposed method requires repeated training on both P and Q for multiple models. I wonder if this could be a potential reason for experimenting on a small dataset. It might be helpful if the authors could further comment on the running costs of the method for larger datasets.

On the related work side, there have recently been many attempts to directly estimate the model performance with a (drifted) unlabelled test set, which is undoubtedly closely associated with this paper. (https://arxiv.org/abs/2202.05834) provides a nice list of some of these methods. In particular, (https://proceedings.mlr.press/v48/platanios16.html) also make use of multiple models' predictions to infer the actual error rate of the test set, which shares a strong link to the proposed method in this paper. It is therefore suggested to justify further the proposed method compared to the aforementioned related work.

Novelty (3/5) The novelty is reasonable as the authors proposed an interesting statistical test on changing performance with the test set. As mentioned above, the problem has many existing attempts, and the idea of using model disagreements can be shared by existing work.

Reproducibility (5/5): This paper provides an excellent example of reproducible research. Both the methods and experiments are described in detail. The source code is also available for additional in-depth investigation.

**Strength And Weaknesses:**

(+) The problem of detecting performance change with an unlabelled dataset is a practical question for many ML applications.

(+) The proposed framework of training disagreeing models and using the model predictions to build test statistics provides an interesting development to the research problem.

(-) While this paper has good coverage of related work on drift (covariate shift) detection, the coverage of methods that estimate the model performance on unlabelled datasets is limited. This limited coverage could be crucial as some methods use similar ideas of modelling the disagreement among multiple models and can be used directly to solve the targeted research problem. (See below for details.)

(-) Some details of the paper might need additional justification, such as the objective learning function for model disagreements, the proposed test statistics, and baseline approaches. (See below for details).

**Summary Of The Paper:**

This paper considers the problem of detecting covariate shifts where a trained model would perform differently. In particular, the authors propose a framework that prepares new models that disagree with the base model and then uses the disagreement rates (or prediction entropy) to build the statistical test.

**Summary Of The Review:**

While the proposed framework provides interesting solutions for training different models and building the statistical tests, this paper might require further work on involving and discussing critical related work before getting published as high-impact research.


=======After author feedback============

Thank the authors again for the extensive feedback. The added explanation and experiments regarding performance estimation are a nice addition to the paper. I have decided to change the score to 6 as a result, but I think some systematic refinement on the paper would make it much stronger (i.e. the disagreement loss, and connections to performance estimators).

---

> ### Author Response · Authors · 2022-11-13
> **Response from Authors (1/2)**
>
> Thank you very much for your review, we have used your feedback to improve our manuscript.
>
> Here is the summary of changes we’ve made to the manuscript
>
> - Clarity - We have fixed several issues regarding notational clarity throughout the paper and listed them in the common response.
> - Additional experiments - We are in the progress of evaluating the use case of methods to predict unlabeled test performance for detecting covariate shift. We have also cited the approaches you have pointed us to in our related work.
> - Additional theory - We develop a notion of proper scoring for disagreement and prove that the disagreement cross entropy satisfies it.
>
> ### Novelty and Comparing to Performance Estimation with an Unlabelled Test Set
>
> > *there have recently been many attempts to directly estimate the model performance with a (drifted) unlabelled test set, which is undoubtedly closely associated with this paper.*
> >
>
> **There is no prior work that uses our idea of training models to disagree on unlabelled data to derive an exact two-sample test with a bounded significance level.**
>
> > The novelty is reasonable as the authors proposed an interesting statistical test on changing performance with the test set. As mentioned above, the problem has many existing attempts, and the idea of using model disagreements can be shared by existing work.
> >
>
> First, we’d like to note that model our experiments after recent methods on using prediction (in these cases via deep learning) for two sample testing (e.g. [https://openreview.net/forum?id=KB5onONJIAU](https://openreview.net/forum?id=KB5onONJIAU), [https://arxiv.org/abs/2002.09116](https://arxiv.org/abs/2002.09116), [https://arxiv.org/abs/1810.11953](https://arxiv.org/abs/1810.11953), [https://arxiv.org/abs/1610.06545](https://arxiv.org/abs/1610.06545))
>
> Thank you for sharing the reference to [Yu et al](https://arxiv.org/abs/2202.05834). This work focuses on estimating performance on unlabelled data. Although similar on the surface, this is not the same problem as identifying harmful covariate shift because the risk on unlabelled data can vary (relative to the training set) due to model misspecification, covariate, label or concept shift.
>
> One way to compare the two is to artificially create a two-sample test (doing so is already well beyond the method presented in) using the algorithm in [Yu et al]. We will attempt a permutation testing approach based on this idea. **We are working to add the results of such a comparison to the paper. For now, we have cited them in related work.**
>
> ### Disagreement Cross-Entropy and Proper Scoring
>
> > *authors provide … theoretical properties (i.e. proper scoring rules) … of the proposed loss*
> >
>
> **Thank you for this suggestion. We build a new definition of *a proper scoring rule for disagreement* and prove that the DCE satisfies this definition.** The proof can be found in Appendix D.1 along with other details on the definition, stability and implementation of the DCE. We now reference this appendix section right after the definition of DCE on page 4.
>
> ### Architecture Choice and deep-MMD Featurizer
>
> > *deep-MMD performed poorly in detecting drifts that even affect model performance. […] I realized that the experiments are performed with quite a limited sample size. In this case, I might further consider reducing the complexity of the projection net*
> >
>
> We’d like to highlight that one of the key advantages of our work is that it no longer necessitates an architecture search for shift detection — we can simply leverage a pertained model for the prediction task.
>
> For deep MMD/CTST we use the original architecture proposed by the authors in our experiments. The default choice is a shallow 4 layer CNN for image datasets and a 4 layer MLP for tabular data. **However, following your suggestion we have run an experiment with CIFAR10/10.1 using a 2 layer CNN. We find that the test power on the 10 sample case drops from 24% to 18% likely because such a small network is not able to capture meaningful features from CIFAR 10.**

---

> > ### Author Response · Authors · 2022-11-13
> > **Response from Authors (2/2)**
> >
> > ### Dataset Size and Runtime Complexity
> >
> > > *it is clear that the proposed method requires repeated training on both P and Q for multiple models. I wonder if this could be a potential reason for experimenting on a small dataset.*
> > >
> >
> > Our real motivation for running on small datasets was the desire in many high-risk domains to be able to detect shift *********quickly.********* This is why our work focuses on the small samples in Q. **We also highlight that many methods in the literature ignore good detection in the small sample regime when evaluating their proposed methods and ours is among the first to focus on it (motivated by the predictive tasks laid out by our clinical collaborators).**
> >
> > A more expansive version of your question is whether Detectron **********fails********** when data is large.
> >
> > In our runtime study (Figure 4) we show that we can achieve a shift detection performance of nearly 100% at a sample size of 100 for the CIFAR10/10.1 benchmark by finetuning a model for only 10 batches of size 512. Put differently, as the number of samples in Q increases, we can get the same performance as the original Detectron by training classifiers to agree on subsamples of P.
> >
> > ---
> > Please let us know if you have any additional thoughts/suggestions.

---

> > > ### Comment · Reviewer_En7A · 2022-11-17
> > > **Regarding proper scoring rules**
> > >
> > > Thank the authors for the extensive feedback.
> > >
> > > At the moment, I am not sure if I am convinced by the newly added definition of (proper scoring rule for disagreement).
> > >
> > > The original concept of PSRs defines a set of loss functions that ensures the probability estimates are equal to the actual probability measures at expectation. It aims to obtain statistical estimates where an unknown ground-truth value is well-defined (i.e. probability measure). However, I am not sure if there is a well-accepted quantity that defines a ground-truth value for model disagreements. Therefore, I don't think it is suitable to refer to the current definition of properness like in PSRs.
> > >
> > > Furthermore, the definition of PSRs includes a set of properties (e.g. convexity, divergence) that are shared by different loss functions (e.g. cross-entropy, squared loss). At the moment, the added discussion is mainly about the proposed measure of DCE, which is not really on a generalised level.
> > >
> > > To wrap up on this specific point, while I appreciate the added discussion on DCE, I don't think the current paper is in the position to define (1) properness for disagreements and (2) a set of loss functions that meet the properness. Therefore I suggest the authors consider renaming the section and contributing to something else, such as "properties of the proposed DCE measure".

---

> > > > ### Author Response · Authors · 2022-11-17
> > > > **Response from Authors**
> > > >
> > > > Thank you for your feedback. We share the sentiment that more work must be done before a fully rigorous notion of a proper score for disagreement can be defined and acknowledge that this is not the objective of our work where we focus primarily on a practical method for identifying covariate shift with small samples. **We have revised our definition to instead define a property *a valid disagreement loss function* and updated the claims in the text accordingly.**
> > > >
> > > > On your point that:
> > > >
> > > > > *I am not sure if there is a well-accepted quantity that defines a ground-truth value for model disagreements.*
> > > > >
> > > >
> > > > We agree with this point fully as it motivates our original definition of PSR for disagreement. Given that there is no “ground truth” for disagreement we instead define a set of distributions $Q$ where the argmax prediction disagrees with a given target $y$. We then prove that the DCE is **valid** in the sense that there is always a distribution $q\in Q$ such that the score awarded for predicting $q$ is lower than any score given to a prediction $p\not\in Q$.

---

> ### Author Response · Authors · 2022-11-17
> **Comparison to Projection Norm [Yu el al 2022]**
>
> As suggested, we compare our method to the contemporary work of Yu et al (****Predicting Out-of-Distribution Error with the Projection Norm****). While this method is not designed for covariate shift detection we use a similar approach of statistical bootstrapping and permutation testing to derive power estimates at a $5\%$ significance level. We ran this experiment on the CIFAR 10/10.1 benchmark used in our experiments. We use the author's [original source code](https://github.com/yaodongyu/ProjNorm) modifying only the dataset, pretrained model, and sample sizes to match our work.
>
> We can observe that the projection norm is suitable score for use as a test statistic for covariate shift, but it does not outperform Detectron in the small sample regime.
>
> |  | $\|Q\|=10$ | $\|Q\|=20$ | $\|Q\|=50$ |
> | --- | --- | --- | --- |
> | ProjNorm [Yu et al] | $.11\pm .03$ | $.17\pm.04$ | $.39\pm .05$ |
> | Detectron (Entropy) | $\mathbf{.35\pm.05}$ | $\mathbf{.56\pm.05}$ | $\mathbf{.92\pm .03}$ |
>
> As we increase the sample size for the projection norm method we observe a catchup in test power. For sample sizes above 100 Detectron achieves a test power of $1$.
>
> || $\|Q\|=100$ | $\|Q\|=200$ | $\|Q\|=500$ | $\|Q\|=1000$ |
> |---| --- | --- | --- | --- |
> | ProjNorm [Yu et al] | $.48\pm .05$ | $.76\pm .04$ | $.99\pm .01$ | $1.0\pm 0$ |
>
> We are currently adding these results to the manuscript for the other datasets used in our work. Please let us know if you have any other thoughts or concerns.

---

> ### Author Response · Authors · 2022-12-05
> **Follow up Reply from Authors**
>
> Thank you again for your hard work during this review process.
>
> Summarizing the responses below we have
>
> - fixed several points of clarity noted in the text
> - implemented a new baseline for the ProjectionNorm approach [[Yu et al](https://arxiv.org/abs/2202.05834) 2022]
> - provided a further theoretical analysis of our proposed loss function
> - performed a small ablation of architecture size on the Deep MMD test
>
> Please let us know if you are satisfied with the changes we’ve implemented and if you have any more comments/questions.
>
> Thank you again for your help in improving our work!

---

### Official Review · Reviewer_wkvQ · 2022-10-28

**Confidence:** 2
**Correctness:** 2
**Technical Novelty And Significance:** 2
**Empirical Novelty And Significance:** 3
**Recommendation:** 5

**Clarity, Quality, Novelty And Reproducibility:**

Clarity: One of the main problems of this paper is clarity.
 - In Problem Setup of Section 3, $N$ is defined as the number of classes. In Theorem 1 and Algorithm 1, it becomes the sample size of $\mathbf{P}$ and $\mathbf{Q}$, respectively.
 - In Problem Setup of Section 3, $\mathbf{P}$ is defined as a set of $n$ labeled samples, but in equation (3), it is a set of input data.
 - In the Constrained Disagreement Classifier, $\mathcal{P}$ is not defined.
 - PQ learning part (page 4): $\epsilon$ and $\delta$ are not defined, and it is not clear the definition of $rej_h (x)$ and $err_h (\tilde{x})$. How are they expressed as mathematical equations? What types of out-of-distributions are considered?

Reproducibility:
 - Code is submitted.



**Strength And Weaknesses:**

- Strength
  - The idea of using an ensemble of models that maximize out-of-domain disagreement makes sense.
  - Related work section fairly covers literature in the field.
  - Extensive experiments are performed.

- Weakness
  - The paper critically lacks clarity and is not mathematically rigorous. It really causes lots of confusion. Please see Clarity, Quality, Novelty, And Reproducibility.
  - Implication of Theorem 1 is not clear. As far as I understand, it is almost impossible to know when the following condition holds: the rate which $g$ disagrees with $f$ on $n$ unseen samples from $\mathcal{Q}$ is greater than that from $n$ unseen samples from $\mathcal{P}$ w.p. greater than $p^{*} = (1-4^{-n} \binom{2n}{n} )/2$. Given that the probability is derived from the expectation of such random experiments, we might need multiple sets of $n$ unseen samples, which is not realistic.

- Question:
  - Theorem 1: do you have an insight into why $p^*$ is independent of $N$? Does it imply that this theorem holds even $N=1$? It is not intuitive...


**Summary Of The Paper:**

This work develops a new hypothesis testing procedure for identifying harmful covariate shifts. To identify harmful covariate shifts, the authors propose to use disagreement cross entropy for multiple models and present Detectron (Disagreement) based on a permutation test.

**Summary Of The Review:**

The submitted paper considers an interesting problem, and the proposed approach makes sense at a high level. However, it lacks clarity which causes much confusion, so I think it is not ready for publication.

---

> ### Author Response · Authors · 2022-11-19
> **Response from Authors**
>
> ### On Theorem 1
>
> The purpose of theorem 1 is to connect the problem of covariate shift detection to model disagreement in a limited/simplified setting.  The main contribution of our work is a learning algorithm and associated statistical test that overcomes the limitation of the theorem.
>
> > *Theorem 1: do you have an insight into why is independent of the number of training samples? Does it imply that this theorem holds even? It is not intuitive...*
> >
>
> **We note that this theorem is not a generalization bound or a convergence proof of our training algorithm, it is a statistical result intended to motivate our practical approach for detecting covariate shift using disagreement.**
>
> The theorem states that **if there exists** a classifier $g$ that is observed to have specific properties on unseen data then there must be covariate shift $\mathcal{P\neq Q}$. We do not assume any requirements on the classifiers $f$ and $g$ in terms of their training performance in the theorem implication and hence the result must be independent of the training set size. **To make this more clear we have removed the mention of the training set size from the theorem statement.**
>
> > *As far as I understand, it is almost impossible to know when the following condition holds: the rate at which $g$ disagrees with $f$ on $n$ unseen samples …*
> >
>
> This is correct, we cannot in general know when the probability bound in Eq 10 will hold; this motivates the need for a statistical test. In Appendix C we conclude that “This result naturally lends itself to identifying $\mathcal{P ≠ Q}$ by rejecting an exact statistical hypothesis that $R_Q = R_P$ in favour of the alternative $R_Q > R_P$”.
>
> We have also remarked in the main text and proved in Appendix C Th. 2  that there is an equivalent Bayesian formulation of Th. 1 which places a uniform prior over the “true” disagreement rates for $\mathcal{P}$ and $\mathcal{Q}$ allowing us to compute an exact probability that the disagreement rates differ using observations from finite samples. You may refer to the post theorem discussion and associated Figure 7 to understand this result without the need to go through the full proof.
>
> > Given that the probability is derived from the expectation of such random experiments, we might need multiple sets of unseen samples, which is not realistic.
> >
>
> Our method requires statistical bootstrapping of distribution $\mathcal{P}$ to compute a quantile of the ID disagreement rate $\phi_{\mathbf{P}}$ and hence will require a significantly larger set of unseen data from $\mathcal{P}$ compared to $\mathcal{Q}$. We argue that this is not infeasible in the small sample regime where we may wish to detect distribution shift on as few as $10$ samples from $\mathcal{Q}$. In such a case a practitioner would only need to hold out $\approx 50-100$ IID samples from the original training set, which they ideally should have already done to create their IID test set.
>
> ### **Clarity**
>
> Thank you for these notes, we have addressed all points of clarity and listed changes in the common response.

---

> > ### Comment · Reviewer_wkvQ · 2022-12-04
> > **Thank you for the response.**
> >
> > First, I thank the authors for the response. The revised version addressed many concerns about the overall clarity of the paper. As for Theorem 1, I agree with the authors that it is not a generalization bound or a convergence proof of the training algorithm, but it is not fully convincing if it really motivates the proposed approach for detecting covariate shifts using disagreement. Also, I am less sure if Theorem 1 is necessarily useful. Although I increase my score from 3 to 5, I believe realistic theoretical results can improve this paper more.

---

> > > ### Author Response · Authors · 2022-12-05
> > > **Reply from Authors**
> > >
> > > Thank you for your hard work in this review process. We are happy to have addressed your concerns about clarity throughout the text and very much appreciate your change in score.
> > >
> > > We wish to highlight further that our work is primarily about building a practical method to detect distribution shifts in the small sample regime. Our theoretical result is useful and sound for the regime where we have unseen samples, however, we argue that this is not usually feasible in the very small sample regime (e.g. there are not enough samples to use some for training and others for testing).  Hence, while the result of the theorem is not *directly* useful for our proposed method, it is useful to pose disagreement as a test statistic for covariate shift and its theoretical drawbacks motivate the transductive approach proposed in our work. We believe that our proposed approach is intuitively motivated by the underlying learning principles (e.g. PQ learning, generalization and overfitting) as well as backed up with rigorous experiments. We also mention in our conclusion that further exploration of the theoretical connection with generalization and model complexity is an interesting path for future work.
> > >
> > > Thank you again for helping improve the quality of our work!

---

### Official Review · Reviewer_DJjN · 2022-11-02

**Confidence:** 3
**Correctness:** 2
**Technical Novelty And Significance:** 3
**Empirical Novelty And Significance:** 2
**Recommendation:** 6

**Clarity, Quality, Novelty And Reproducibility:**

The novelty and Quality are fair. This work contributes some new ideas. It has minor technical flaws and some typos. The notations should be improved. For example, the notation $[ \cdot ]$ in Eq. (2) should be an indicator function and there is no definition of $\hat y_c$.
The clarity is poor. The content should be carefully reorganized.


**Strength And Weaknesses:**

**Strength:**
1. The authors propose disagreement cross-entropy to improve the optimization procedure of the PQ-learning problem.
2. This work develops a hypothesis testing method to detect covariate shifts and the detection principle is consistent with the common theoretical understanding that extrapolation increases the variance of outputs.
3. Experiments show that the proposed method is sample efficient.

**Weakness:**

1. The motivation is unclear. In Section 3, the generalization set $\mathcal{R}$ is a set of distributions. Is it possible that $\mathcal{P}$'s support set overlaps with $\mathcal{Q}$'s support set? If $\mathcal{Q}$'s support set is a subset of $\mathcal{P}$'s support set, how to understand the harmful covariate shift? Can you disentangle the effect of correlation shift ( the relation between the inputs and outputs $P(Y|X)$ ) and the effect of harmful covariate shift?
2. Need more analysis on the utility of disagreement-cross-entropy. In Eq. (3), if we remove the disagreement-cross-entropy loss, how does the performance of Detectron change? Does the learning procedure of $g$ change the correlation between $Y$ and the features in $f(X)$?
3. What are the requirements for the pre-trained model $f$? Is $f$ required to achieve good prediction accuracy? How does the proposed detection method perform if the ID data is ImageNet?
4. If the pre-trained model $f$ is over-parameterized, can we force the disagreement rate on ID data to be 0%? Then how to implement Algorithm 1?
5. Theorem 1: "… If the rate … greater than … there must be covariate shift." This is already a hard threshold. Why do you still consider a hypothesis test approach?
6. Can we consider the task in this paper as an OOD detection task with auxiliary data? If so, a comparison with recent OOD Detection methods is required, e.g. POEM.




**Summary Of The Paper:**

The authors propose a learning-based testing method to identify distribution shifts (a new domain) with access to finite samples of the test domain. The authors define harmful covariate shift (HCS) that may weaken the generalization of a predictive model, and constrained disagreement classifiers (CDC) that performs consistently on ID data and inconsistently on OOD data. Then they detect HCS by the inconsistency of the outputs of the constrained disagreement classifiers. Theoretical analysis shows that the disagreement rate of CDCs can be used to detect covariate shifts. The proposed algorithm uses the empirical distribution of the CDCs on the training data to determine the reject region.

**Summary Of The Review:**

I recommend marginal rejection. Overall, this is a potentially promising work. But I do not think the current version is ready for publication.

---

> ### Author Response · Authors · 2022-11-19
> **Response from Authors (1/2)**
>
> ### Novelty and Quality/Clarity
>
> > The novelty and Quality are fair. This work contributes some new ideas. It has minor technical flaws and some typos. The notations should be improved. For example, the notation $[⋅]$ in Eq. (2) should be an indicator function and there is no definition of $y_c$.
> >
>
> **We have fixed several points of clarity and listed them in the common response.**
>
> ### Motivation for Generalization Sets
>
> > The motivation is unclear. In Section 3, the generalization set $\mathcal{R}$ is a set of distributions. Is it possible that $\mathcal{P}$'s support set overlaps with $\mathcal{Q}$'s support set? If $\mathcal{Q}$'s support set is a subset of $\mathcal{P}$'s support set, how to understand the harmful covariate shift?
> >
>
> We would like to note that identifying the support of distributions for high dimensional data is not feasible when only observing finite samples.
>
> In our work we define $\mathcal{R}$ as an abstract set of distributions for which a classifier $f$ achieves a given generalization error. We argue that in general $\mathcal{R}$ contains more than just what be likely under $\mathcal{P}$ as most models leverage or learn certain invariances that allow them to generalize (for more details and a simple example with rotations on MNSIT see Appendix G).
>
> If $\mathcal{Q}$ is among the generalization set $\mathcal{R}$ then we say there is no harmful shift and vice-versa. We propose the constrained disagreement objective as a practical tool for identifying if a distribution $\mathcal{Q}$ is part of $\mathcal{R}$ given only a finite set of sample drawn from it.
>
> This concept is explained at the end of the subsection on Harmful Covariate Shift and addressed in more detail in Appendices G and H.
>
> > Can you disentangle the effect of correlation shift ( the relation between the inputs and outputs  $P(Y|X)$) and the effect of harmful covariate shift?
> >
>
> We only assume access to unlabeled data from the test distribution hence we cannot make any claims related to correlation/concept shift. This point is stated explicitly in the introduction and problem setup.
>
> ### Utility of the Disagreement Cross Entropy
>
> > Need more analysis on the utility of disagreement-cross-entropy.
> >
>
> **We have added further theoretical analysis on the disagreement cross entropy (DCE) to Appendix D.1** where we introduce a new definition of a “a valid loss function for disagreement” and prove that the DCE satisfies it. We also added a proof (Appendix D.2) to show that the DCE presents as a special case of a more intuitive generalized disagreement loss explained in the main text and elaborated on in Appendix D.2.
>
> > In Eq. (3), *if we remove the disagreement-cross-entropy loss, how does the performance of Detectron change?*
> >
>
> Removing the disagreement term from Equation 3 would have the effect of ignoring all unlabeled data making our method equivalent to fine-tuning a model using the only original training objective and training set. Related to this approach we consider using deep ensembles (rows 3 and 4 of Table 1) where instead of CDCs we use the disagreement/entropy of an ensemble of models trained only on $\mathbf{P}$ with different random seeds.
>
> **We have rewritten Equation 3 (main training objective) to make it more clear that the DCE is the only component that leads to interaction with the unlabeled data** $\mathbf{Q}$**.**
>
> > *Does the learning procedure of* $g$ *change the correlation between $Y$ and the features in $f(X)$?*
> >
>
> The learning procedure of the constrained disagreement classifier **$g$ does not update the base model $f$ (see Appendix A.2 Algorithm 3 for constrained disagreement). Importantly we also monitor $g$ during training to make sure it matches the in-distribution performance of $f$ i.e. fitting the disagreement objective $\tilde{\ell}$ does not make $g$ comprise on in domain accuracy
>
> Please let us know if we have addressed your question on this point.

---

> > ### Author Response · Authors · 2022-11-19
> > **Response from Authors (2/2)**
> >
> > ### Requirements on Pertained Models
> >
> > > *What are the requirements for the pre-trained model? Is required to achieve good prediction accuracy? How does the proposed detection method perform if the ID data is ImageNet?*
> > >
> >
> > In our work, we choose models with high in distribution generalization error (see Appendix F.2 Table 2  for a list of models and associated in and out of distribution performance). We consider the problem of auditing a model that has been deployed for a real task, hence there is an implicit assumption that the performance is sufficiently suitable for the domain (e.g. the model is not random).
> >
> > **We have added mention to the conclusion that precise relationships between model complexity, generalization error, and test power are interesting directions for future work. “**Appendix I” in the original manuscript contains a discussion related to the interaction of shift detection and model complexity (see a summary of this section in the response to Reviewer 4).
> >
> > We have established the validity of our method on three datasets spanning different data types(tabular, images) across predictive models that include neural networks and random forests. We have not performed experiments on ImageNet as they are costly to run and seldom used in related work on the identification of dataset shift due to the lack of natural notions of shift in Imagenet.
> >
> > ### Over-parameterized models
> >
> > > *If the pre-trained model is over-parameterized, can we force the disagreement rate on ID data to be 0%? Then how to implement Algorithm 1?*
> > >
> >
> > With overparameterized models the CDC training objective will eventually be fully satisfied as a consequence of overfitting (e.g. 100% agreement on training data and 100% on unlabeled data no-mater if it’s ID or OOD) this would not lead to an informative test. When dealing with overparameterized models it is important to take common precautions against overfitting used in regular supervised learning (e.g. augmentation, early stopping…); this point is noted in the text right after equation 3 and discussed in the experimental details section of the appendix.
> >
> > **We have added a new Appendix section D.3 to specifically note the consequences of overparameterization/overfitting and discuss how simple early stopping measures can be employed to resolve them without the need for additional data. The section also includes an example taken from an experiment with CIFAR10/10.1.**
> >
> > We would like to further highlight that this issue is not unique to our method, other work that takes a learning approach to covariate shift detection (e.g. H-divergence, CTST, deep-MMD) may also suffer from loss of statistical significance due to overparameterization+overfitting.
> >
> > ### On Theorem 1
> >
> > > *Theorem 1: "… If the rate … greater than … there must be covariate shift." This is already a hard threshold. Why do you still consider a hypothesis test approach?*
> > >
> >
> > Theorem 1 establishes the statistical convergence properties of a test for covariate shift based on disagreement computed on ************unseen data. H************owever, after the Theorem statement we note that applying it directly is impractical in the small sample regime since we do not have enough samples to hold out for unbiased testing. This is why we take a transductive approach and compute the disagreement rate directly on the **observed data** used to train the CDC model itself thus eliminating the need for unseen samples. The approach can no longer leverage the unbiased guarantees we present hence requiring the use of an exact statistical test (e.g. permutation test).
> >
> > Please see additional commentary on the implications of Theorem 1 in the response to reviewer 2.
> >
> > ### Relation to OODD with Auxiliary Data
> >
> > > Can we consider the task in this paper as an OOD detection task with auxiliary data? If so, a comparison with recent OOD Detection methods is required, e.g. POEM.
> > >
> >
> > **In this work we do not assume access to auxiliary outliers at training time.** We assume only that a model has been trained on a labelled dataset $\\mathbf{P}=\\{(x_i,y_i)\\}_{i=1}^n$ and that we wish to asses a new unlabeled dataset $\\mathbf{Q}=\\{\\tilde{x}_i\\}_\{i=1\}^m$ for covariate shift at a fixed significance level. Designing better two sample tests for settings where auxiliary outliers are available is a potential area for future work.

---

> ### Author Response · Authors · 2022-12-05
> **Follow up Reply From Authors**
>
> Thank you again for your hard work during this review process.
>
> Summarizing the responses below we have
>
> - fixed several points of clarity noted in the text
> - provided a further theoretical analysis of our proposed loss function
> - added a new appendix on the potential consequences of overparameterization/overfitting and they should be mitigated
> - discussed relations between model complexity and test power as an avenue for future work
> - highlighted the notable differences between our setting and that of ODD with auxiliary data
>
> Please let us know if you are satisfied with the changes we’ve implemented and if you have any more comments/questions.
>
> Thank you again for your help in improving our work!

---

> > ### Comment · Reviewer_DJjN · 2022-12-08
> > **After Rebuttal**
> >
> > Thanks for the rebuttal. I have checked the revised paper and I believe this addresses most of my concerns about your paper. I will increase my score to 6.
> >
> > Further comments：
> > - For the generalization set, the definition is still ambiguous. If the "generalization" here refers to the consistency of the prediction results and does not disentangle the covariate shift or correlation shift, we can directly discuss this problem on the support set rather than a set of distributions.
> > - Overparameterization leads to a non-informative test. However, overparameterization may not lead to poor generalization.
> > -  The unlabeled dataset **Q** is the auxiliary data.

---

> > > ### Author Response · Authors · 2022-12-08
> > > **Addressing Remaining Points**
> > >
> > > We thank you again for your time and dedication in reviewing our work and additionally for engaging in this follow-up discussion.
> > >
> > > We briefly highlight a response to each of your three remaining points.
> > >
> > > 1) **Detectron does not disentangle covariate shift and correlation shift**
> > >
> > > As **we operate in the setting where we only observe unlabeled data** at test time we cannot detect shifts of the form $p_S(Y|X)\neq p_T(Y|X)$ as $Y$ is unobserved. The generalization set is a heuristic notion that refers to the set of distributions only over $X$ for which a model has *generalized* due to invariances learned by the model. An implicit assumption for generalization outside of the training domain to be possible is the lack of arbitrary correlation shifts. Any subsequent updates of this work will state this assumption explicitly when defining the generalization set to avoid any ambiguity.  Thank you for addressing this point of clarity.
> > >
> > > 2) **Detectron does work with overparameterized models**
> > >
> > > We agree that overparameterization is an essential component of the success of ML and wish to highlight that **our method does indeed work and is mainly designed for use with overparameterized models**. Evidence can be seen in this [figure](https://s3.us-west-2.amazonaws.com/secure.notion-static.com/47850786-8230-4f7f-bb19-1fb1dae467de/Untitled.png?X-Amz-Algorithm=AWS4-HMAC-SHA256&X-Amz-Content-Sha256=UNSIGNED-PAYLOAD&X-Amz-Credential=AKIAT73L2G45EIPT3X45%2F20221208%2Fus-west-2%2Fs3%2Faws4_request&X-Amz-Date=20221208T164529Z&X-Amz-Expires=86400&X-Amz-Signature=706f2b8e144bf76786cedc95613180ec571b83f40a9ed2b25dd8ec4e55322674&X-Amz-SignedHeaders=host&response-content-disposition=filename%3D"Untitled.png"&x-id=GetObject) (figure 9 appendix D3 in the manuscript). The figure shows that when running the detectron using an IID unlabeled set $\mathbf{P}^\star$ the disagreement rate grows significantly slower compared to running it on a non-exchangeable set $\mathbf{Q}$. While overparameterization implies that the training loss eventually goes to zero, (equivalently in this case the disagreement rate of a CDC eventually goes to 1) the power of the Detectron comes from its ability to exploit the dynamics of the proposed constrained disagreement algorithm for detecting distribution shift.
> > >
> > > 3) **Difference in the setting of our work with** **OODD with auxiliary data**
> > >
> > > In the work you have mentioned on OODD with auxiliary data [POEM (Ming et al 2022)] the authors assume access to an auxiliary set of outliers available at training time. They go on to introduce a “training objective that incorporates the sampled outliers for model regularization”.
> > >
> > > In contrast **we do not assume access to a set of outliers at training time**. Instead, we consider the task of auditing a pre-trained model **in deployment where we only observe unlabeled samples** but do not know if they are outliers or not. We would also like to further point out that no others work that we are aware of in the space of high-dimensional two-sample testing draw any comparisons with OODD with Auxiliary Data.
> > >
> > > Please let us know if this distinction is clear or if you believe we are missing additional details.

---

### Author Response · Authors · 2022-11-13
**Common Response to all Reviewers**

We would like to thank all of the reviewers for their detailed feedback. Here we highlight common responses and main changes to the manuscript.

### Disagreement Learning

In our work, we proposed the disagreement cross entropy (DCE) as a simple loss function for training gradient-based learning models to disagree with a given target label and output high entropy predictions. We have added additional commentary on the DCE in Appendix D.1 including defining a new and sensible notion of a “valid loss function for disagreement” and proving that the DCE is indeed valid under our definition. We have also added a simple proof to show that the generalized disagreement loss (appendix D.2) is equivalent to the DCE as a special case.

### Addressing Limitations

We have extended our conclusion to address the limitations of our method when using models from a non-sufficiently rich function class and have referenced a more in-depth discussion with specific examples in Appendix I. We also discuss the promising avenue of developing a relationship between model generalization and shift detection performance.

### Notation/Clarity

All points of notational clarity pointed out by the reviewers as well have been fixed throughout the text

- $N$ is now only used to represent the number of classes, $n$ the sample size of $\mathbf{P}$, and $m$ the sample size of $\mathbf{Q}$. Theorem 1, the first paragraph of page 5, algorithm 1, and the experimental setup have been adjusted.
- The less familiar Iverson bracket $[\cdot ]$ has been replaced with the standard binary indicator $1_{(\cdot)}$
- $\hat{y}_c$ in Eq 2, page 3 has been clarified as the $c^{\text{th}}$ element of the vector $\hat{y}$
- Eq 3 (main loss function) has been rewritten to improve the clarity
- More specifics on the notation used in the definition $PQ$ learning, including precise mathematical definitions of the error and rejection rate of selective classifiers are added to Appendix A.1 and referenced in the “limitations of PQ learning” section of the main text
- The definition of the constrained disagreement classifier property is reworded to say that $g$ but achieves similar performance on unseen samples drawn identically from the same distribution as dataset $\mathbf{P}$ (i.e. iid test samples)

---

### Decision · Program_Chairs · 2023-01-20

**Decision:**

Accept: poster

**Justification For Why Not Higher Score:**

This paper has both theoretical and experimental components. I feel that theoretical contributions are fair but not extensive.

**Justification For Why Not Lower Score:**

The reviewers agreed that the proposed approach that relies on disagreement of two models is practical and novel. The reviewers find this aspect unique. It is worth publishing.

**Metareview: Summary, Strengths And Weaknesses:**

The paper proposes a method to detect covariate shift by training two classifiers to agree on a distribution P and disagree on another distribution Q. The disagreement rate of the two classifiers is used to detect covariate shift as part of a statistical test.

The discussion phase was fruitful with meaningful exchanges from all reviewers that result in better understanding on the reviewers’ side and better submission compared to the initial submission. Some important negative points raised and **resolved** were

* Lack of analysis of the disagreement cross entropy (DJjN); -> The authors added an analysis to the appendix
* Lack of clarity and justification on some technical parts (wkvQ, En7A) -> revised and the reviewers’ concerns were resolved
* Corner cases not discussed (e.g., models underfitting and disagree) -> resolved by the rebuttal.

The reviewers agreed that the proposed approach that relies on disagreement of two models is practical and novel. The proposal is theoretically grounded (DJjN), and interesting (En7A). The submission covers related work very well (En7A, wkvQ), and has extensive experiments (wkvQ). Concerns were addressed satisfactorily after the rebuttal.

Recommendation: accept.


**Note From Pc:**

if the above contains the word "oral" or "spotlight" please see: "oral" presentation means -> notable-top-5% and "spotlight" means -> notable-top-25%. As stated in our emails, we are disassociating presentation type from AC recommendations

**Summary Of Ac-Reviewer Meeting:**

N/A